# ON INFORMATION-THEORETIC MEASURES OF PREDICTIVE UNCERTAINTY

## ABSTRACT

Reliable estimation of predictive uncertainty is crucial for machine learning applications, particularly in high-stakes scenarios where hedging against risks is essential. Despite its significance, a consensus on the correct measurement of predictive uncertainty remains elusive. In this work, we return to first principles to develop a fundamental framework of information-theoretic predictive uncertainty measures. Our proposed framework categorizes predictive uncertainty measures according to two factors: **(I)** The predicting model **(II)** The approximation of the true predictive distribution. Examining all possible combinations of these two factors, we derive a set of predictive uncertainty measures that includes both known and newly introduced ones. We empirically evaluate these measures in typical uncertainty estimation settings, such as misclassification detection, selective prediction, and out-of-distribution detection. The results show that no single measure is universal, but the effectiveness depends on the specific setting. Thus, our work provides clarity about the appropriateness of predictive uncertainty measures by clarifying their implicit assumptions and relationships.

## 1 INTRODUCTION

Integrating machine learning models into high-stakes scenarios, such as autonomous driving or managing critical healthcare systems, introduces substantial risks. To hedge against these risks, we need to quantify the uncertainty associated with each prediction to prevent models from making decisions that carry both significant risk and uncertainty. In such cases, it is better to defer uncertain decisions to human experts or opt for a safer, though potentially less advantageous, alternative decision. Consequently, it is vital to employ reliable measures of predictive uncertainty and provide estimates for them when implementing machine learning models for decision making in high-stakes applications.

The entropy of the posterior predictive distribution has become the standard information-theoretic measure to assess predictive uncertainty (Houlsby et al., 2011; Gal, 2016; Depeweg et al., 2018; Smith and Gal, 2018; Mukhoti et al., 2023). Despite its widespread use, this measure has drawn criticism (Malinin and Gales, 2021; Wimmer et al., 2023), prompting the proposal of alternative information-theoretic measures (Malinin and Gales, 2021; Schweighofer et al., 2023b;a; Kotelevskii and Panov, 2024; Hofman et al., 2024b). The relationship between those measures is still not well understood, although their similarities suggest that they are special cases of a more general formulation.

We show that all these measures are approximations of the cross-entropy between the predicting model and the true model. However, since the true model is not known in general, this fundamental measure is intractable to compute directly. By considering different assumptions about the predicting model and approximations of the true model, we develop a framework to categorize information-theoretic measures of predictive uncertainty. Our framework includes existing measures, introduces new ones, and clarifies the relationship between these measures. Furthermore, our empirical analysis reveals that the effectiveness of different measures varies depending on the task and the posterior sampling method used. In sum, our contributions are as follows:

1. We introduce a unifying framework to categorize measures of predictive uncertainty according to assumptions about the predicting model and how the true model is approximated. This framework not only encompasses existing measures but also suggests new ones and clarifies their relationship.

2. We derive our framework from first principles, based on the cross-entropy between the predicting model and the true model as the fundamental yet intractable measure of predictive uncertainty.

3. We empirically evaluate these measures across various typical uncertainty estimation tasks and show that their effectiveness depends on the setting and the posterior sampling method used.

## 2 QUANTIFYING PREDICTIVE UNCERTAINTY

We consider the canonical classification setting with inputs $\boldsymbol{x} \in \mathbb{R}^D$ and targets $y \in \mathcal{Y}$, where $\mathcal{Y}$ is the set of all $K$ possible targets. The dataset $\mathcal{D}$ is given, sampled i.i.d according to the data generating distribution. We consider deep neural networks as a class of probabilistic models that map an input $\boldsymbol{x}$ to the $K - 1$ dimensional probability simplex $\Delta^{K-1} = \{\boldsymbol{\theta} \in \mathbb{R}^K \mid \theta_k \geq 0 \, \forall k, \sum_{k=1}^{K} \theta_k = 1\}$. This mapping is defined as $f_{\boldsymbol{w}} : \mathbb{R}^D \to \Delta^{K-1}$ for a model with parameters $\boldsymbol{w}$. The output of this mapping defines the distribution parameters of a categorical distribution, in the following referred to as the model's predictive distribution $p(y \mid \boldsymbol{x}, \boldsymbol{w}) = \mathrm{Cat}(y; f_{\boldsymbol{w}}(\boldsymbol{x})) = \mathrm{Cat}(y; \boldsymbol{\theta})$.

The predictive distribution of a probabilistic model represents the uncertainty inherent in its predictions. When the probability mass is uniformly distributed across all possible outcomes, it denotes complete uncertainty about the prediction, whereas concentration on a single class indicates complete certainty. If we have access to the true data-generating model, denoted by parameters $\boldsymbol{w}^*$, the predictive distribution $p(y \mid \boldsymbol{x}, \boldsymbol{w}^*)$ captures the inherent and irreducible uncertainty in the prediction, often referred to as *aleatoric uncertainty* (AU) (Gal, 2016; Kendall and Gal, 2017). This assumes that the chosen model class can accurately represent the true predictive distribution, thus $p(y \mid \boldsymbol{x}) = p(y \mid \boldsymbol{x}, \boldsymbol{w}^*)$, which is a common and often necessary assumption (Hüllermeier and Waegeman, 2021). The information-theoretic entropy $\mathrm{H}(\cdot)$ of the true predictive distribution is a natural and universally accepted measure of aleatoric uncertainty, defined as

$$\mathrm{H}(p(y \mid \boldsymbol{x}, \boldsymbol{w}^*)) := -\sum_{k=1}^{K} p(y = k \mid \boldsymbol{x}, \boldsymbol{w}^*) \log p(y = k \mid \boldsymbol{x}, \boldsymbol{w}^*) . \tag{1}$$

However, we generally don't know the true model and have to choose parameters $\boldsymbol{w}$ out of all possible ones. Consequently, uncertainty arises due to the lack of knowledge about the true parameters of the model. This is called *epistemic uncertainty* (EU) (Apostolakis, 1990; Helton, 1993; 1997; Gal, 2016). An effective measure of predictive uncertainty should be consistent with Eq. (1) and capture both AU and EU, usually assumed to sum up to a total predictive uncertainty (TU).

### 2.1 STANDARD MEASURE: ENTROPY OF THE POSTERIOR PREDICTIVE DISTRIBUTION

Given a dataset $\mathcal{D}$ and prior $p(\boldsymbol{w})$ on the model parameters, Bayes' theorem yields the posterior distribution $p(\boldsymbol{w} \mid \mathcal{D})$. The posterior distribution denotes the probability that the parameters $\boldsymbol{w}$ match the true parameters $\boldsymbol{w}^*$ of the model that generated the dataset $\mathcal{D}$. Instead of committing to a single model, the posterior distribution allows marginalizing over all possible models, which is known as Bayesian model averaging. This gives rise to the posterior predictive distribution

$$p(y \mid \boldsymbol{x}, \mathcal{D}) = \mathrm{E}_{p(\boldsymbol{w}|\mathcal{D})} [p(y \mid \boldsymbol{x}, \boldsymbol{w})] . \tag{2}$$

The entropy of the posterior predictive distribution is the currently most widely accepted approach to measure predictive uncertainty (Houlsby et al., 2011; Gal, 2016; Depeweg et al., 2018; Smith and Gal, 2018; Hüllermeier and Waegeman, 2021; Mukhoti et al., 2023). According to a well-known result from information theory (Cover and Thomas, 2006), this entropy can be additively decomposed into the conditional entropy and the mutual information I between $y$ and $\boldsymbol{w}$:

$$\mathrm{H}(p(y \mid \boldsymbol{x}, \mathcal{D})) = \underbrace{\mathrm{E}_{p(\boldsymbol{w}|\mathcal{D})} [\mathrm{H}(p(y \mid \boldsymbol{x}, \boldsymbol{w}))]}_{\text{aleatoric}} + \underbrace{\mathrm{I}(p(y, \boldsymbol{w} \mid \boldsymbol{x}, \mathcal{D}))}_{\text{epistemic}} . \tag{3}$$

Furthermore, Eq. (3) is equivalent to a decomposition of expected cross-entropy $\mathrm{CE}(\cdot \, ; \, \cdot)$ into conditional entropy and expected KL-divergence $\mathrm{KL}(\cdot \parallel \cdot)$ (Schweighofer et al., 2023b;a):

$$\mathrm{E}_{p(\boldsymbol{w}|\mathcal{D})} [\mathrm{CE}(p(y \mid \boldsymbol{x}, \boldsymbol{w}) \, ; \, p(y \mid \boldsymbol{x}, \mathcal{D}))] \tag{4}$$
$$= \underbrace{\mathrm{E}_{p(\boldsymbol{w}|\mathcal{D})} [\mathrm{H}(p(y \mid \boldsymbol{x}, \boldsymbol{w}))]}_{\text{aleatoric}} + \underbrace{\mathrm{E}_{p(\boldsymbol{w}|\mathcal{D})} [\mathrm{KL}(p(y \mid \boldsymbol{x}, \boldsymbol{w}) \parallel p(y \mid \boldsymbol{x}, \mathcal{D}))]}_{\text{epistemic}} .$$

If the parameters of the true model are known, EU vanishes and Eq. (3) as well as Eq. (4) simplify to Eq. (1), thus are consistent with it. However, the entropy of the posterior predictive distribution has been found to be inadequate for specific scenarios, such as autoregressive predictions (Malinin and Gales, 2021) or for a given predicting model (Schweighofer et al., 2023b) and was criticised on grounds of not fulfilling certain expected theoretical properties (Wimmer et al., 2023). In response, alternative information-theoretic measures have been introduced (Malinin and Gales, 2021; Schweighofer et al., 2023b;a; Kotelevskii and Panov, 2024; Hofman et al., 2024b). Although the relationship between these measures is not well understood, their structure similar to Eq. (4) suggests a connection between them. We next propose a fundamental, though generally intractable, predictive uncertainty measure, where all of these measures are special cases under specific assumptions.

## 2.2 Proposed Measure: Cross-Entropy between Selected and True Distribution

An effective measure of total predictive uncertainty should incorporate epistemic uncertainty and be consistent with Eq. (1). Considering this, we propose to measure predictive uncertainty with the cross-entropy between the predictive distributions of a selected predicting model and the true model. Let $p(y \mid \boldsymbol{x}, \cdot)$ be the predictive distribution of any selected model for some new input $\boldsymbol{x}$, which we will refer to as *predicting model*. We will examine different cases for the predicting model later; for now, it suffices to consider it to be a specific model with parameters $\boldsymbol{w}$. The cross-entropy between the predictive distributions of the predicting model and the true model is given by

$$
\mathrm{CE}(p(y \mid \boldsymbol{x}, \cdot) \, ; \, p(y \mid \boldsymbol{x}, \boldsymbol{w}^*)) := -\sum_{k=1}^{K} p(y = k \mid \boldsymbol{x}, \cdot) \log p(y = k \mid \boldsymbol{x}, \boldsymbol{w}^*) \tag{5}
$$

$$
= \underbrace{\mathrm{H}(p(y \mid \boldsymbol{x}, \cdot))}_{\text{aleatoric}} + \underbrace{\mathrm{KL}(p(y \mid \boldsymbol{x}, \cdot) \, \| \, p(y \mid \boldsymbol{x}, \boldsymbol{w}^*))}_{\text{epistemic}} \, .
$$

If the predictive distribution of the predicting model is equal to the predictive distribution of the true model, the epistemic component is zero by definition and Eq. (5) simplifies to Eq. (1). Thus, as expected, if the parameters of the true model are known, the epistemic uncertainty vanishes. Eq. (5) is a fundamental, though generally intractable, measure of predictive uncertainty. To obtain tractable measures, assumptions about the predicting model and about how to approximate the true model are necessary. This gives rise to our framework, which we introduce in detail in Sec. 3. As an example, comparing the standard measure in Eq. (4) with our proposed measure in Eq. (5), we observe that for the standard measure, the predicting model is any model according to its posterior probability, and the posterior predictive distribution is considered to be the true predictive distribution.

**Interpretation of aleatoric and epistemic uncertainty.** An important distinction compared to previous work is in our interpretation of aleatoric and epistemic uncertainty, which aligns with the understanding of Apostolakis (1990); Helton (1993; 1997) as follows. The aleatoric component is not generally understood as a property of the true predictive distribution, but of the selected predicting model used to make a prediction. Thus, it is the uncertainty that arises due to predicting with the selected probabilistic model. The epistemic component is defined as the additional uncertainty due to predicting with the selected predicting model instead of the true model. Thus, it is the additional uncertainty that arises due to selecting a model from the given model class.

## 3 Proposed Framework of Predictive Uncertainty Measures

Our proposed measure of predictive uncertainty (Eq. (5)) allows for different assumptions about (I) the selected predicting model and (II) how to approximate the true model. For both of them, we consider three different assumptions. This yields nine different measures of predictive uncertainty within our proposed framework. An overview of all measures is given in Tab. 1, summarizing the total predictive uncertainties as well as their aleatoric and epistemic components.

### (A,B,C): Predicting Model

The most obvious choice of a predicting model is (A) a pre-selected given model with parameters $\boldsymbol{w}$. This is the standard case in machine learning, where model parameters are selected, e.g. by maximizing the likelihood on the training dataset or downloaded from a model hub.

Table 1: **Our proposed framework of information-theoretic measures of predictive uncertainty.** Each measure denotes a different instantiation of the fundamental measure given by Eq. (5) for different assumptions about the predicting model and how the true model is approximated. For brevity, we define $p_{\boldsymbol{w}} := p(y \mid \boldsymbol{x}, \boldsymbol{w})$, $p_{\mathcal{D}} := p(y \mid \boldsymbol{x}, \mathcal{D})$, and $\mathrm{E}_{\boldsymbol{w}} := \mathrm{E}_{p(\boldsymbol{w}|\mathcal{D})}$ (the same for $\tilde{\boldsymbol{w}}$). Expressions with the same cell coloring are equivalent to each other. Each measure of total predictive uncertainty additively decomposes into an aleatoric and epistemic component by $\mathrm{CE}(p \; ; \; q) = \mathrm{H}(p) + \mathrm{KL}(p \parallel q)$.

| | Predicting model | | Approximation of the true predictive distribution | | |
| --- | --- | --- | --- | --- | --- |
| | | | (1) $\tilde{\boldsymbol{w}}$ | (2) $\mathrm{E}_{\tilde{\boldsymbol{w}}}$ | (3) $\tilde{\boldsymbol{w}} \sim p(\tilde{\boldsymbol{w}} \mid \mathcal{D})$ |
| TU | (A) $\boldsymbol{w}$ | | $\mathrm{CE}(p_{\boldsymbol{w}} \; ; \; p_{\tilde{\boldsymbol{w}}})$ | $\mathrm{CE}(p_{\boldsymbol{w}} \; ; \; p_{\mathcal{D}})$ | $\mathrm{E}_{\tilde{\boldsymbol{w}}}\left[\mathrm{CE}(p_{\boldsymbol{w}} \; ; \; p_{\tilde{\boldsymbol{w}}})\right]$ |
| | (B) $\mathrm{E}_{\boldsymbol{w}}$ | | $\mathrm{CE}(p_{\mathcal{D}} \; ; \; p_{\tilde{\boldsymbol{w}}})$ | $\mathrm{CE}(p_{\mathcal{D}} \; ; \; p_{\mathcal{D}})$ | $\mathrm{E}_{\tilde{\boldsymbol{w}}}\left[\mathrm{CE}(p_{\mathcal{D}} \; ; \; p_{\tilde{\boldsymbol{w}}})\right]$ |
| | (C) $\boldsymbol{w} \sim p(\boldsymbol{w} \mid \mathcal{D})$ | | $\mathrm{E}_{\boldsymbol{w}}\left[\mathrm{CE}(p_{\boldsymbol{w}} \; ; \; p_{\tilde{\boldsymbol{w}}})\right]$ | $\mathrm{E}_{\boldsymbol{w}}\left[\mathrm{CE}(p_{\boldsymbol{w}} \; ; \; p_{\mathcal{D}})\right]$ | $\mathrm{E}_{\boldsymbol{w}}\left[\mathrm{E}_{\tilde{\boldsymbol{w}}}\left[\mathrm{CE}(p_{\boldsymbol{w}} \; ; \; p_{\tilde{\boldsymbol{w}}})\right]\right]$ |
| AU | (A) $\boldsymbol{w}$ | | $\mathrm{H}(p_{\boldsymbol{w}})$ | $\mathrm{H}(p_{\boldsymbol{w}})$ | $\mathrm{H}(p_{\boldsymbol{w}})$ |
| | (B) $\mathrm{E}_{\boldsymbol{w}}$ | | $\mathrm{H}(p_{\mathcal{D}})$ | $\mathrm{H}(p_{\mathcal{D}})$ | $\mathrm{H}(p_{\mathcal{D}})$ |
| | (C) $\boldsymbol{w} \sim p(\boldsymbol{w} \mid \mathcal{D})$ | | $\mathrm{E}_{\boldsymbol{w}}\left[\mathrm{H}(p_{\boldsymbol{w}})\right]$ | $\mathrm{E}_{\boldsymbol{w}}\left[\mathrm{H}(p_{\boldsymbol{w}})\right]$ | $\mathrm{E}_{\boldsymbol{w}}\left[\mathrm{H}(p_{\boldsymbol{w}})\right]$ |
| EU | (A) $\boldsymbol{w}$ | | $\mathrm{KL}(p_{\boldsymbol{w}} \parallel p_{\tilde{\boldsymbol{w}}})$ | $\mathrm{KL}(p_{\boldsymbol{w}} \parallel p_{\mathcal{D}})$ | $\mathrm{E}_{\tilde{\boldsymbol{w}}}\left[\mathrm{KL}(p_{\boldsymbol{w}} \parallel p_{\tilde{\boldsymbol{w}}})\right]$ |
| | (B) $\mathrm{E}_{\boldsymbol{w}}$ | | $\mathrm{KL}(p_{\mathcal{D}} \parallel p_{\tilde{\boldsymbol{w}}})$ | $\mathrm{KL}(p_{\mathcal{D}} \parallel p_{\mathcal{D}})^{\,0}$ | $\mathrm{E}_{\tilde{\boldsymbol{w}}}\left[\mathrm{KL}(p_{\mathcal{D}} \parallel p_{\tilde{\boldsymbol{w}}})\right]$ |
| | (C) $\boldsymbol{w} \sim p(\boldsymbol{w} \mid \mathcal{D})$ | | $\mathrm{E}_{\boldsymbol{w}}\left[\mathrm{KL}(p_{\boldsymbol{w}} \parallel p_{\tilde{\boldsymbol{w}}})\right]$ | $\mathrm{E}_{\boldsymbol{w}}\left[\mathrm{KL}(p_{\boldsymbol{w}} \parallel p_{\mathcal{D}})\right]$ | $\mathrm{E}_{\boldsymbol{w}}\left[\mathrm{E}_{\tilde{\boldsymbol{w}}}\left[\mathrm{KL}(p_{\boldsymbol{w}} \parallel p_{\tilde{\boldsymbol{w}}})\right]\right]$ |

Another widely used method is (B) the Bayesian model average (c.f. Eq. (2)). Here, instead of predicting with a single model, the predictive distribution is marginalized over all possible models according to their posterior probability. In practice, exact marginalization is often intractable and therefore approximated by posterior sampling.

Finally, it is possible to (C) consider every possible model as the predicting model, weighted by their posterior probabilities. This might seem counterintuitive, as it means that the predicting model is not fixed but is sampled anew for each prediction. Nevertheless, the aleatoric component of the resulting uncertainty measures, denoted $\mathrm{E}_{p(\boldsymbol{w}|\mathcal{D})}\left[\mathrm{H}(p(y \mid \boldsymbol{x}, \boldsymbol{w}))\right]$, is the best approximation of the aleatoric uncertainty under the true model for a given posterior distribution. However, as pointed out by Wimmer et al. (2023), it is neither a lower nor an upper bound on the aleatoric uncertainty under the true model and is highly dependent on the posterior distribution.

(1,2,3): APPROXIMATION OF THE TRUE PREDICTIVE DISTRIBUTION

The simplest but probably biased choice to approximate the true predictive distribution is (1) the predictive distribution under a single given model with parameters $\tilde{\boldsymbol{w}}$. Although this might be a poor approximation, it might be the only feasible choice in specific settings. For example, it is used in speculative decoding (Stern et al., 2018; Leviathan et al., 2023), where a small model is used to predict and its predictive distribution is compared against a large model that serves as the ground truth.

Another possibility is to use (2) the posterior predictive distribution as an approximation of the true predictive distribution. Although intuitively appealing, Schweighofer et al. (2023a) criticized this as there is no guarantee that these distributions coincide, even for a perfect estimate of the posterior predictive distribution. Furthermore, there are degenerate cases where the posterior predictive distribution can't be represented by any model with non-vanishing posterior probability. However, it is often a well performing approximation empirically for expressive models such as neural networks. Additionally, (2) is the only option that guarantees finite EU and as a result TU.

Finally, perhaps the most intuitive solution is to consider (3) all possible models according to their posterior probability. Any model could be the true model according to its posterior distribution.

Therefore, we should consider the mismatch between the predictive distribution of the selected predicting model and all other models, weighted by their posterior probability.

### 3.1 Relationships between Measures

Importantly, the aleatoric components of the uncertainty measures depend only on the predicting model and do not depend on the approximation of the true predictive distribution. Thus, they are the same for cases (1), (2) and (3). Furthermore, the aleatoric component of case (B) is an upper bound of the aleatoric component of case (C), i.e. $\mathrm{H}(p(y \mid \boldsymbol{x}, \mathcal{D})) \geq \mathrm{E}_{p(\boldsymbol{w}|\mathcal{D})}[\mathrm{H}(p(y \mid \boldsymbol{x}, \boldsymbol{w}))]$, which directly follows from Eq. (3) as the mutual information is non-negative.

Due to the linearity in the first argument of the cross-entropy, the total uncertainties for cases (B) and (C) are equal. Furthermore, as already discussed, the aleatoric components for cases (B) and (C) differ by the mutual information $\mathrm{E}_{p(\boldsymbol{w}|\mathcal{D})}[\mathrm{KL}(p(y \mid \boldsymbol{x}, \boldsymbol{w}) \parallel p(y \mid \boldsymbol{x}, \mathcal{D}))]$. Therefore, the epistemic components for cases (B) and (C) also differ by this factor. This is trivial to see for cases (B2) and (C2), where the epistemic component of case (B2) cancels to zero and the epistemic component of case (C2) is the mutual information. For cases (B3) and (C3), this was already mentioned by (Malinin and Gales, 2021) and a proof was given by (Schweighofer et al., 2023a), which we include for completeness in Sec. A.1 in the appendix, together with a version for cases (B1) and (C1).

### 3.2 Categorization of Previously Known Measures

The standard measure (Eq. (4)) introduced by Houlsby et al. (2011) and popularized, for instance, by Gal (2016); Depeweg et al. (2018); Smith and Gal (2018) is the measure (C2). In the context of autoregressive predictions, Malinin and Gales (2021) introduced measure (B3), due to the feasibility of a Monte Carlo (MC) approximation compared to the standard measure (C2). Schweighofer et al. (2023b) introduced measure (A3) together with a posterior sampling algorithm that is explicitly taylored to this measure. Schweighofer et al. (2023a) introduced measure (C3) as an improvement over the standard measure (C2) for certain settings. Hofman et al. (2024b) also derived measure (C3) for the logarithmic strictly proper scoring rule (log score). Furthermore, Kotelevskii and Panov (2024) discussed measures (B2), (B3), (C2) and (C3) as Bayesian approximations under the log score. Our work thus generalizes and gives a new perspective on those measures.

## 4 Related Work

**Measures of predictive uncertainty.** The currently most widely used information-theoretic measure of predictive uncertainty is the entropy of the posterior predictive distribution (Eq.(3)). In Sec. (3.2), we discuss the relationship of previous work based on this measure and our proposed framework. However, there are also other measures of predictive uncertainty, not based on information-theoretic quantities. Depeweg et al. (2018) introduced variance-based measures, based on the law of total variance. This perspective was recently developed further for specific settings (Duan et al., 2024; Sale et al., 2023b). Furthermore, Sale et al. (2024b) introduced label-wise measures of predictive uncertainty, formulating both information-theoretic and variance-based measures. Another idea recently proposed by Sale et al. (2024a) is quantifying uncertainty through distances to reference (second-order) distributions for TU, AU, and EU, respectively, which represent complete certainty. Thus, the higher the distance from the reference distribution, the more uncertain the prediction. All measures discussed so far operate on a distributional representation of uncertainty. Orthogonal to that, there are also set-based approaches (Hüllermeier et al., 2022; Sale et al., 2023a; Hofman et al., 2024a).

**Posterior sampling methods.** All measures proposed by our framework, except (A1), contain a posterior expectation. Those are generally approximated by sampling models according to the posterior distribution. An obvious choice are MCMC algorithms, for example HMC (Neal, 1995; Neal et al., 2011), which has recently been investigated on modern neural network architectures (Izmailov et al., 2021). Scaling HMC to large datasets and architectures is computationally costly. However, more efficient approximate variants using stochastic gradients are also available (Welling and Teh, 2011; Chen et al., 2014; Zhang et al., 2020). Furthermore, it is possible to learn a simpler variational distribution that approximates the posterior distribution. Widely known examples are the mean-field approach of Blundell et al. (2015) or MC Dropout (Gal and Ghahramani, 2016). Another approach is the Laplace approximation (MacKay, 1992) around a maximum a posteriori (MAP) model (Ritter

et al., 2018; Daxberger et al., 2021). A commonly used approximation to the Bayesian ideal are Deep Ensembles (Lakshminarayanan et al., 2017), which despite their algorithmic simplicity are widely recognized to provide high-quality samples (Wilson and Izmailov, 2020; Izmailov et al., 2021). Furthermore, Schweighofer et al. (2023b) introduced adversarial models to explicitly search for models with a large contribution to approximating expectations of the epistemic component for case (3). For a more extensive overview, see, e.g. Gawlikowski et al. (2023) or Papamarkou et al. (2024).

## 5 EXPERIMENTS

In this section, we evaluate the performance of the proposed measures across various experimental scenarios that leverage uncertainty, including tasks like misclassification detection, selective prediction, and out-of-distribution (OOD) detection. In addition, we assess the impact of various posterior sampling methods, which is a crucial factor in real-world applications. We do not intend to identify the optimal measure for a specific task or posterior sampling method; all of them can be evaluated, and the best chosen in practice. Our primary aim is to deepen the understanding of our proposed framework.

**Datasets.** Our experiments are performed on the CIFAR10/100 (Krizhevsky and Hinton, 2009), SVHN (Netzer et al., 2011), Tiny-ImageNet (TIN) (Le and Yang, 2015) and LSUN (Yu et al., 2015) datasets. For TIN, we resize the inputs to 32x32 to match the other datasets. We train models on all datasets except LSUN, which is used solely as an OOD dataset.

**Models and training.** We used three different model architectures for our experiments: ResNet-18 (He et al., 2016), DenseNet-169 (Huang et al., 2017) and RegNet-Y 800MF (Radosavovic et al., 2020). Individual models were trained for 100 epochs using SGD with momentum of 0.9 with a batch size of 256 and an initial learning rate of 1e-2. Furthermore, a standard combination of linear (from factor 1 to 0.1) and cosine annealing schedulers was used. The results discussed in the main paper are obtained using ResNet-18 as the model architecture. For the other two architectures, results are provided in Sec. B.4 of the appendix. Those are consistent with the findings presented in the main paper.

**Predictive uncertainty measures.** We consider all measures proposed by our framework, c.f. Tab. 1. For example, the (total) measure (A1) is referred to as TU (A1), its aleatoric component as AU (A) and its epistemic component as EU (A1). Here, AU (A) is used over AU (A1) to emphasize the independence of the aleatoric component from the approximation of the true model.

**Posterior sampling methods.** We consider three methods to sample models according to the posterior $p(\boldsymbol{w} \mid \mathcal{D})$, Deep Ensembles (DE) (Lakshminarayanan et al., 2017), Laplace Approximation (LA) (MacKay, 1992) on the last layer with Kronecker-factored approximate curvature (Ritter et al., 2018) using the implementation of Daxberger et al. (2021) and MC Dropout (MCD) (Gal and Ghahramani, 2016). Those samples are used to approximate posterior expectations. For example, the posterior predictive distribution given by Eq. (2) is approximated by

$$p(y \mid \boldsymbol{x}, \mathcal{D}) \approx \frac{1}{N} \sum_{n=1}^{N} p(y \mid \boldsymbol{x}, \boldsymbol{w}_n), \qquad \boldsymbol{w}_n \sim p(\boldsymbol{w} \mid \mathcal{D}) \qquad (6)$$

with $N$ samples. Posterior expectations within the proposed measures are approximated in the same way. We provide formulas for the MC approximations for all measures as well as their aleatoric and epistemic components in Sec. A.2 in the appendix. For all three methods, we sample 10 models for the MC approximations of the uncertainty measures. Measures based on a single model (combinations with (A) and (1)) use the first member of the ensemble for DE, the maximum a posteriori (MAP) model for LA, and the model without dropout activated for MCD.

There is a distinction between multi- and single-basin posterior sampling techniques (Wilson and Izmailov, 2020), sometimes also referred to as multi- and single-mode approaches (Hoffmann and Elster, 2021). We refer to them as global and local posterior sampling techniques for simplicity. In this categorization, DE is a global method, while LA and MCD are local methods (Fort et al., 2019). We hypothesize that different methods for posterior sampling have a strong impact on which uncertainty measure performs well empirically, especially given whether they are global or local methods.

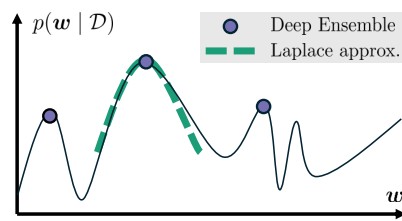

Figure 1: Posterior sampling methods.

## 5.1 CHARACTERISTICS OF POSTERIOR SAMPLES

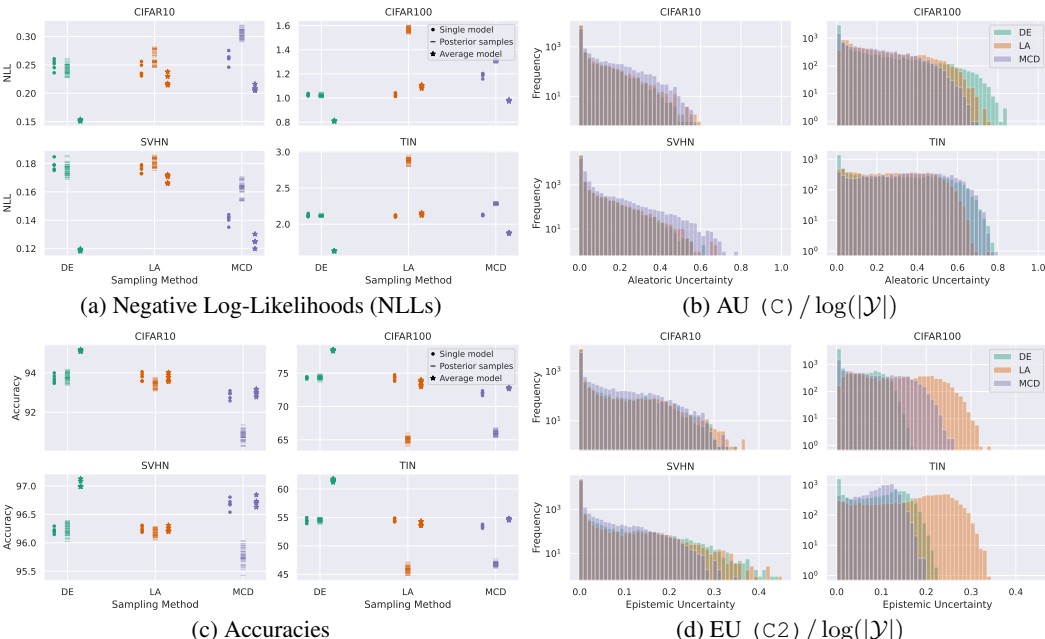

(a) Negative Log-Likelihoods (NLLs)

(b) AU $(\mathtt{C}) / \log(|\mathcal{Y}|)$

(c) Accuracies

(d) EU $(\mathtt{C2}) / \log(|\mathcal{Y}|)$

Figure 2: **Comparison of posterior sampling methods.** Results are obtained on the test split of the respective dataset. We compare the NLLs (a) and accuracies (c) for different models obtained through DE, LA and MCD. Similarly, (b) the normalized AU $(\mathtt{C})$ and (d) the normalized EU $(\mathtt{C2})$ are shown per sampling method. All three methods yield similar results for CIFAR10 and SVHN, but differ greatly on CIFAR100 and TIN. Models sampled using LA have higher NLL and lower accuracy. Furthermore, they lead to higher epistemic uncertainty and lack predictions with very low aleatoric uncertainty. Additionally, the average model does not improve over the single model in terms of NLL and accuracy for those two datasets. The results in (a) and (c) show single models, posterior samples and average models of five independent runs, those in (b) and (d) uncertainties for a single run.

To better understand the performance of different posterior sampling methods, we examine the characteristics of their sampled models. The results in Fig. 2 show that these methods perform differently across datasets. For the global sampling method DE, the average model consistently outperforms individual sampled models with a lower negative log-likelihood (NLL) and higher accuracy across all datasets. In contrast, for local sampling methods LA and MCD, individual sampled models exhibit higher NLL than both the single model and the average model. Additionally, the accuracy of individual sampled models is lower than that of the single model. Specifically, for MCD, the single model's accuracy is comparable to the average model, while for LA, the single model's accuracy exceeds that of the average model.

We further analyze the predictive uncertainties estimated by different posterior sampling methods using measure $(\mathtt{C2})$, which incorporates posterior samples and is upper-bounded. To ensure comparability across datasets, we normalize the uncertainties by the maximal predictive uncertainty TU $(\mathtt{C2})$, equal to the entropy of the uniform distribution $\log(|\mathcal{Y}|)$. The results in Fig. 2b and d show that these methods yield similar distributions of uncertainties for CIFAR10 and SVHN. However, for CIFAR100 and TIN, DE exhibits many more datapoints with very low EU and AU.

## 5.2 MISCLASSIFICATION DETECTION

We sampled models on the CIFAR10/100, SVHN and TIN datasets using DE, LA and MCD and obtain predictions on the respective test datasets. This was done with (i) the single model, (ii) the average model and (iii) some model according to the posterior distribution to investigate the impact of aligning the measure of uncertainty with the predicting model - $(\mathtt{A})$ for (i), $(\mathtt{B})$ for (ii) and $(\mathtt{C})$ for (iii). The single model for (i) is a random but fixed model for DE, the MAP model for LA, and the model without dropout for MCD. The average model for (ii) is defined by Eq. (6), averaging over all sampled

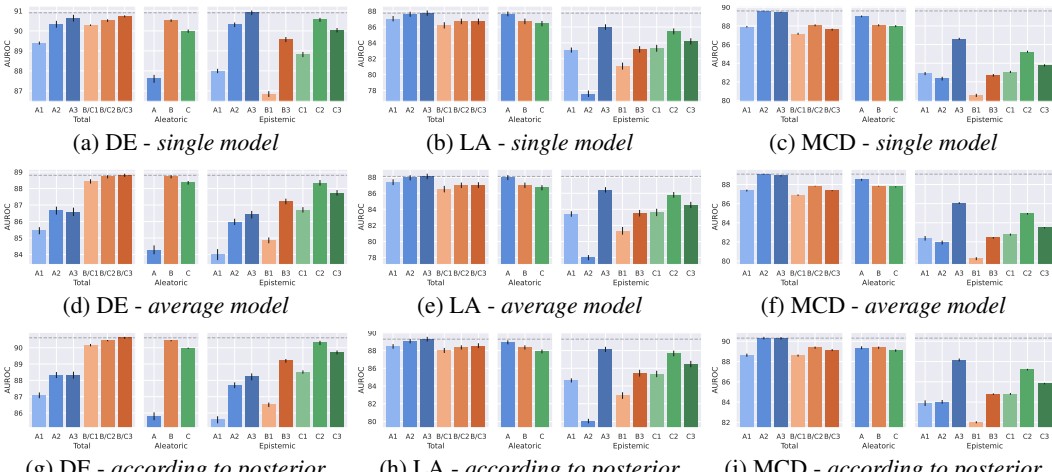

Figure 3: **Misclassification detection under different predicting models.** AUROC for distinguishing correct from incorrect predictions under different predicting models, using the different proposed measures of uncertainty as score. The global method DE performs best for EU (A3) when predicting with the single model. Otherwise it performs best for TU (B/C3). The local methods LA and MCD perform best for TU (A2) and TU (A3) no matter the predicting model. AUROCs are averages over all datasets with statistics over five independent runs.

models. For (iii), one model from the sampled models was randomly selected for each prediction. Note that (iii) does not make much sense in practice, as individual models are performing worse than the average model in terms of accuracy for all considered methods, the same as a single model for DE and worse than the single model for LA and MCD (see Fig. 2c). We compare the AUROC for distinguishing between correctly and incorrectly predicted datapoints for the different proposed measures of predictive uncertainty as scoring functions. Alternative measures commonly used to evaluate misclassification such as AUPR or FPR@TPR95 were also considered. Those induced the same ordering of uncertainty measures, thus we report the AUROC throughout all experiments.

The results are given in Fig. 3. We average over the four considered datasets and report means and standard deviations over five independent runs. The results for individual datasets are reported in Fig. 10 - Fig. 12 in the appendix. To detect misclassifications of (i) the single model, EU (A3) performs best (Fig. 3a). However, when predicting with (ii) the average or (iii) a model according to the posterior, TU (B/C3) performs best (Fig. 3d,g). For the local method LA, TU (A3) performs best regardless of the predicting model (Fig. 3b,e,h). The same effect is observed for MCD (Fig. 3c,f,i), yet TU (A2) slightly outperforms TU (A3) in this case. We hypothesize that this effect occurs because the local methods fail to provide high-quality samples for some datasets, resulting in high variance of posterior estimates and thus low accuracy. In sum, we find that TU (A2) and TU (A3) perform well for local posterior sampling methods regardless of the predicting model, but for global posterior sampling methods aligning the measure with the predicting model makes a strong difference.

## 5.3 SELECTIVE PREDICTION

Another commonly considered task is selective prediction, where the model's predictions are limited to a specific subset, and its performance is evaluated on that subset. The setup in this experiment is identical to the misclassification setup. We evaluated the accuracy for a subset of predictions of (i) the single model, (ii) the average model, and (iii) a model according to the posterior distribution. Subsets between 50% of the most certain datapoints and the entire dataset were considered. The area under the accuracy retention curve (AUARC) was used as performance measure to compare the efficacy of uncertainty measures to provide a ranking to select those subsets.

We focus on (i) the single model and (ii) the average model using DE with the results given in Fig. 4. Additional results are provided in Sec. B.2 in the appendix. The results show a similar picture as for misclassification detection, where the optimal measure depends on the model used for prediction. For (i) the single model, TU (A3) performs best, while for (ii) the average model, TU (B/C3) performs best. Again, the best measures are those aligned to the predicting model.

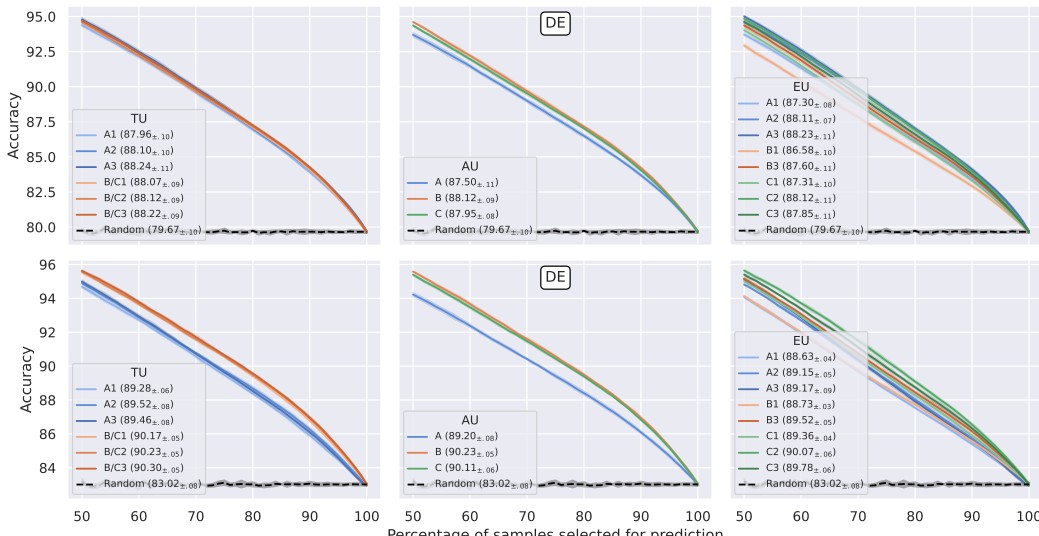

Figure 4: **Selective prediction for single and average model for DE.** Accuracies per fraction of datapoints the single model (top row), the average model (bottom row) predicts on, as well as area under the accuracy retention curve (tabulated in legend) using different predictive uncertainty measures as score. Accuracies are averaged over all datasets with statistics over five independent runs.

## 5.4 OOD Detection

We sampled models on CIFAR10/100, SVHN and TIN using DE, LA and MCD. Therefore, we use the respective test dataset as in-distribution (ID) datasets and the test datasets for the others, as well as LSUN, as OOD datasets. OOD detection does not involve a prediction by the model. Thus, it is not possible to align the uncertainty measure with the predicting model, as in misclassification detection and selective prediction. We compare the AUROC for distinguishing between ID and OOD datapoints for each measure within our framework as a scoring function. Alternative commonly used measures such as the AUPR and the FPR@TPR95 were also considered. However, since they induced the same ordering of measures, we report the AUROC for all OOD detection experiments.

The results are shown in Fig. 5. We observe that throughout all measures, the total and the aleatoric components perform much better than the epistemic components, which is contrary to assumptions commonly formulated in the literature (Mukhoti et al., 2023; Kotelevskii and Panov, 2024). However, this might depend on the datasets. For example, with MCD, the epistemic components perform best on the pairs TIN/CIFAR10 and TIN/CIFAR100 (Fig. 16 c,f). We hypothesize that the strong performance of the aleatoric components is due to the low levels of noise in the considered datasets. Furthermore, for the local method LA, all measures and their aleatoric components perform equally well. For DE and MCD, TU (B/C2) and TU (B/C3) perform best.

## 5.5 Additional Experiments

Our experiments aim to investigate the performance of the proposed framework on a wide range of tasks. Due to space limitations, we moved the following additional experiments to the appendix:

We investigated the performance of the provided measures for detecting distribution shifts on CIFAR10 using the CIFAR10-C (Hendrycks and Dietterich, 2019) dataset. This is a conceptually similar task to OOD detection, as our results provided in Sec. B.5 confirm. We observe that for smaller shifts, the epistemic components perform much better than the others; for larger shifts, this effect vanishes.

Furthermore, we investigated the efficacy of our measures for detecting adversarial examples under FGSM (Goodfellow et al., 2015) and PGD (Madry et al., 2018) attacks. We do not intend to claim any level of adversarial robustness to these attacks, but use them as a tool to understand the behaviour of our measures. The results are discussed in Sec. B.6.

Finally, we conducted active learning experiments on MNIST (Lecun et al., 1998) and FMNIST (Xiao et al., 2017) using DE and MCD as posterior sampling methods for a small convolutional neural

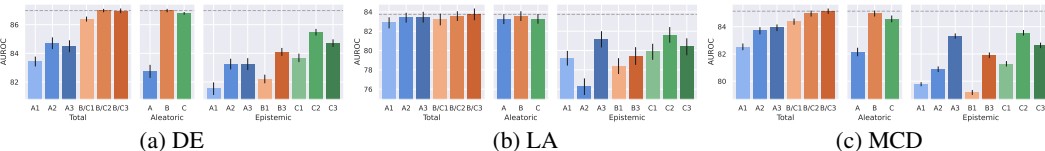

(a) DE     (b) LA     (c) MCD

Figure 5: **OOD detection.** AUROC for distinguishing between ID and OOD datapoints using the different proposed measures of uncertainty as score. For the local method LA, all measures and their aleatoric components perform equally well, for DE and MCD, TU (B/C2) and TU (B/C3) perform best. AUROCs are averaged over all ID / OOD combinations with statistics over five independent runs.

network. Prior work (Gal et al., 2017; Mukhoti et al., 2023) suggests that the optimal acquisition function is a measure of epistemic uncertainty. Our results indicate, that a good acquisition function must capture the mutual information between $y$ and $\boldsymbol{w}$ faithfully rather than the epistemic uncertainty as defined by our framework. The results and an in-depth discussion are given in Sec. B.7.

## 6   CONCLUSION

We have proposed a framework that categorizes measures of predictive uncertainty according to assumptions about the predicting model and how the true model is approximated. This framework has been derived from first principles, based on the cross-entropy between the predicting model and the true model (Eq. (5)). Most importantly, it clarifies the relationships between information-theoretic measures of predictive uncertainty and uncovers their implicit assumptions. Our empirical evaluation shows that the effectiveness of the different measures depends on the task and the posterior sampling method used. As there is no best-performing measure under all conditions, it is crucial to not consider only a single one to benchmark posterior sampling methods for uncertainty quantification.

Our proposed framework for estimating predictive uncertainty requires an approximation of posterior expectations through samples. However, obtaining samples is generally expensive, although many improvements in efficiency have already been made. To avoid this issue, deterministic methods that require only a single forward pass have been proposed. The most prominent directions are evidential models (Sensoy et al., 2018; Amini et al., 2020), prior networks (Malinin and Gales, 2018), as well as feature distance / density based models (Bradshaw et al., 2017; Liu et al., 2020; Van Amersfoort et al., 2020; Mukhoti et al., 2023). Those methods generally utilize different measures of predictive uncertainty than those discussed in this work, and their relation is not thoroughly understood so far.

The information-theoretic framework presented in this work considers individual predictions. However, there is currently a lot of interest around autoregressive predictions, especially for large language models. For such models, uncertainty estimation has been considered as a way to detect hallucinations (Xiao and Wang, 2021). Extending the framework presented in this work to autoregressive predictions comes with a set of challenges (Malinin and Gales, 2021; Kuhn et al., 2023; Aichberger et al., 2024), such as the necessity to sample output sequences to obtain entropy estimates, output sequences of varying length, and semantic equivalences between output sequences. We believe that tackling those issues is an important direction for future work.

## ETHICS STATEMENT

This work considers the foundations of predictive uncertainty estimation. Our primary goal is to increase the robustness and reliability of machine learning models applied to real-world settings. We do not forsee any negative societal impact arising from the findings of this paper and hope to have a positive societal impact by aiding decision making in safety-critical applications.

## REPRODUCIBILITY STATEMENT

We provide a detailed description of our experimental setup, sufficient to be independently reproduced, in Sec. 5. Descriptions for additional experiments are provided in Sec. B.5, Sec. B.6 and Sec. B.7. Furthermore, we provide our implementation as supplementary material and will publicly release the code upon acceptance.

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

## A TECHNICAL DETAILS

### A.1 RELATIONSHIPS BETWEEN EPISTEMIC COMPONENTS

Schweighofer et al. (2023a) proved the relationship that the sum of the epistemic components of C2 and B3 is equivalent to the epistemic component of C3. For completeness, we provide a version of the proof as follows:

$$
\overbrace{\mathrm{E}_{p(\boldsymbol{w}|\mathcal{D})}\left[\mathrm{KL}(p(y\mid\boldsymbol{x},\boldsymbol{w})\parallel p(y\mid\boldsymbol{x},\mathcal{D}))\right]}^{\text{EU (C2) - Mutual Information}} + \overbrace{\mathrm{E}_{p(\tilde{\boldsymbol{w}}|\mathcal{D})}\left[\mathrm{KL}(p(y\mid\boldsymbol{x},\mathcal{D})\parallel p(y\mid\boldsymbol{x},\tilde{\boldsymbol{w}}))\right]}^{\text{EU (B3)}} \tag{7}
$$

$$
= \mathrm{E}_{p(\boldsymbol{w}|\mathcal{D})}\left[\mathrm{E}_{p(y|\boldsymbol{x},\boldsymbol{w})}\left[\log\frac{p(y\mid\boldsymbol{x},\boldsymbol{w})}{p(y\mid\boldsymbol{x},\mathcal{D})}\right]\right] + \mathrm{E}_{p(\tilde{\boldsymbol{w}}|\mathcal{D})}\left[\mathrm{E}_{p(y|\boldsymbol{x},\mathcal{D})}\left[\log\frac{p(y\mid\boldsymbol{x},\mathcal{D})}{p(y\mid\boldsymbol{x},\tilde{\boldsymbol{w}})}\right]\right] \tag{8}
$$

$$
= \mathrm{E}_{p(\boldsymbol{w}|\mathcal{D})}\left[\mathrm{E}_{p(y|\boldsymbol{x},\boldsymbol{w})}\left[\log p(y\mid\boldsymbol{x},\boldsymbol{w})\right] - \mathrm{E}_{p(y|\boldsymbol{x},\boldsymbol{w})}\left[\log p(y\mid\boldsymbol{x},\mathcal{D})\right]\right] + \tag{9}
$$
$$
\mathrm{E}_{p(\tilde{\boldsymbol{w}}|\mathcal{D})}\left[\mathrm{E}_{p(y|\boldsymbol{x},\mathcal{D})}\left[\log p(y\mid\boldsymbol{x},\mathcal{D})\right] - \mathrm{E}_{p(y|\boldsymbol{x},\mathcal{D})}\left[\log p(y\mid\boldsymbol{x},\tilde{\boldsymbol{w}})\right]\right]
$$

$$
= \mathrm{E}_{p(\boldsymbol{w}|\mathcal{D})}\left[\mathrm{E}_{p(y|\boldsymbol{x},\boldsymbol{w})}\left[\log p(y\mid\boldsymbol{x},\boldsymbol{w})\right]\right] - \cancel{\mathrm{E}_{p(y|\boldsymbol{x},\mathcal{D})}\left[\log p(y\mid\boldsymbol{x},\mathcal{D})\right]} + \tag{10}
$$
$$
\cancel{\mathrm{E}_{p(y|\boldsymbol{x},\mathcal{D})}\left[\log p(y\mid\boldsymbol{x},\mathcal{D})\right]} - \mathrm{E}_{p(\tilde{\boldsymbol{w}}|\mathcal{D})}\left[\mathrm{E}_{p(y|\boldsymbol{x},\mathcal{D})}\left[\log p(y\mid\boldsymbol{x},\tilde{\boldsymbol{w}})\right]\right]
$$

$$
= \mathrm{E}_{p(\boldsymbol{w}|\mathcal{D})}\left[\mathrm{E}_{p(y|\boldsymbol{x},\boldsymbol{w})}\left[\log p(y\mid\boldsymbol{x},\boldsymbol{w})\right]\right] - \tag{11}
$$
$$
\mathrm{E}_{p(\tilde{\boldsymbol{w}}|\mathcal{D})}\left[\mathrm{E}_{p(\boldsymbol{w}|\mathcal{D})}\left[\mathrm{E}_{p(y|\boldsymbol{x},\boldsymbol{w})}\left[\log p(y\mid\boldsymbol{x},\tilde{\boldsymbol{w}})\right]\right]\right]
$$

$$
= \mathrm{E}_{p(\boldsymbol{w}|\mathcal{D})}\left[\mathrm{E}_{p(\tilde{\boldsymbol{w}}|\mathcal{D})}\left[\mathrm{E}_{p(y|\boldsymbol{x},\boldsymbol{w})}\left[\log p(y\mid\boldsymbol{x},\boldsymbol{w})\right]\right]\right] - \tag{12}
$$
$$
\mathrm{E}_{p(\boldsymbol{w}|\mathcal{D})}\left[\mathrm{E}_{p(\tilde{\boldsymbol{w}}|\mathcal{D})}\left[\mathrm{E}_{p(y|\boldsymbol{x},\boldsymbol{w})}\left[\log p(y\mid\boldsymbol{x},\tilde{\boldsymbol{w}})\right]\right]\right]
$$

$$
= \mathrm{E}_{p(\boldsymbol{w}|\mathcal{D})}\left[\mathrm{E}_{p(\tilde{\boldsymbol{w}}|\mathcal{D})}\left[\mathrm{E}_{p(y|\boldsymbol{x},\boldsymbol{w})}\left[\log\frac{p(y\mid\boldsymbol{x},\boldsymbol{w})}{p(y\mid\boldsymbol{x},\tilde{\boldsymbol{w}})}\right]\right]\right] \tag{13}
$$

$$
= \underbrace{\mathrm{E}_{p(\boldsymbol{w}|\mathcal{D})}\left[\mathrm{E}_{p(\tilde{\boldsymbol{w}}|\mathcal{D})}\left[\mathrm{KL}(p(y\mid\boldsymbol{x},\boldsymbol{w})\parallel p(y\mid\boldsymbol{x},\tilde{\boldsymbol{w}}))\right]\right]}_{\text{EU (C3)}}, \tag{14}
$$

which is what we wanted to show. The step from (9) to (10) is due to additivity and linearity of expectations. The step from (11) to (12) is due to the fact that we can insert the expectation $\mathrm{E}_{p(\tilde{\boldsymbol{w}}|\mathcal{D})}$ in the first term as it does not depend on $\tilde{\boldsymbol{w}}$ and due to the fact that $p(\tilde{\boldsymbol{w}}\mid\mathcal{D}) = p(\boldsymbol{w}\mid\mathcal{D})$. □

Furthermore, a similar proof can be constructed for EU (C1) = EU (C2) + EU (B1) as follows:

$$
\overbrace{\mathrm{E}_{p(\boldsymbol{w}|\mathcal{D})}\left[\mathrm{KL}(p(y\mid\boldsymbol{x},\boldsymbol{w})\parallel p(y\mid\boldsymbol{x},\mathcal{D}))\right]}^{\text{EU (C2) - Mutual Information}} + \overbrace{\mathrm{KL}(p(y\mid\boldsymbol{x},\mathcal{D})\parallel p(y\mid\boldsymbol{x},\tilde{\boldsymbol{w}}))}^{\text{EU (B1)}} \tag{15}
$$

$$
= \mathrm{E}_{p(\boldsymbol{w}|\mathcal{D})}\left[\mathrm{E}_{p(y|\boldsymbol{x},\boldsymbol{w})}\left[\log\frac{p(y\mid\boldsymbol{x},\boldsymbol{w})}{p(y\mid\boldsymbol{x},\mathcal{D})}\right]\right] + \mathrm{E}_{p(y|\boldsymbol{x},\mathcal{D})}\left[\log\frac{p(y\mid\boldsymbol{x},\mathcal{D})}{p(y\mid\boldsymbol{x},\tilde{\boldsymbol{w}})}\right] \tag{16}
$$

$$
= \mathrm{E}_{p(\boldsymbol{w}|\mathcal{D})}\left[\mathrm{E}_{p(y|\boldsymbol{x},\boldsymbol{w})}\left[\log p(y\mid\boldsymbol{x},\boldsymbol{w})\right] - \mathrm{E}_{p(y|\boldsymbol{x},\boldsymbol{w})}\left[\log p(y\mid\boldsymbol{x},\mathcal{D})\right]\right] + \tag{17}
$$
$$
\mathrm{E}_{p(y|\boldsymbol{x},\mathcal{D})}[\log p(y|\boldsymbol{x},\mathcal{D})] - \mathrm{E}_{p(y|\boldsymbol{x},\mathcal{D})}\left[\log p(y\mid\boldsymbol{x},\tilde{\boldsymbol{w}})\right]
$$

$$
= \mathrm{E}_{p(\boldsymbol{w}|\mathcal{D})}\left[\mathrm{E}_{p(y|\boldsymbol{x},\boldsymbol{w})}\left[\log p(y\mid\boldsymbol{x},\boldsymbol{w})\right]\right] - \cancel{\mathrm{E}_{p(y|\boldsymbol{x},\mathcal{D})}\left[\log p(y\mid\boldsymbol{x},\mathcal{D})\right]} + \tag{18}
$$
$$
\cancel{\mathrm{E}_{p(y|\boldsymbol{x},\mathcal{D})}\left[\log p(y\mid\boldsymbol{x},\mathcal{D})\right]} - \mathrm{E}_{p(y|\boldsymbol{x},\mathcal{D})}\left[\log p(y\mid\boldsymbol{x},\tilde{\boldsymbol{w}})\right]
$$

$$
= \mathrm{E}_{p(\boldsymbol{w}|\mathcal{D})}\left[\mathrm{E}_{p(y|\boldsymbol{x},\boldsymbol{w})}\left[\log p(y\mid\boldsymbol{x},\boldsymbol{w})\right]\right] - \mathrm{E}_{p(\boldsymbol{w}|\mathcal{D})}\left[\mathrm{E}_{p(y|\boldsymbol{x},\boldsymbol{w})}\left[\log p(y\mid\boldsymbol{x},\tilde{\boldsymbol{w}})\right]\right] \tag{19}
$$

$$
= \mathrm{E}_{p(\boldsymbol{w}|\mathcal{D})}\left[\mathrm{E}_{p(y|\boldsymbol{x},\boldsymbol{w})}\left[\log\frac{p(y\mid\boldsymbol{x},\boldsymbol{w})}{p(y\mid\boldsymbol{x},\tilde{\boldsymbol{w}})}\right]\right] \tag{20}
$$

$$
= \underbrace{\mathrm{E}_{p(\boldsymbol{w}|\mathcal{D})}\left[\mathrm{KL}(p(y\mid\boldsymbol{x},\boldsymbol{w})\parallel p(y\mid\boldsymbol{x},\tilde{\boldsymbol{w}}))\right]}_{\text{EU (C1)}}, \tag{21}
$$

which is what we wanted to show. Again, the step from (17) to (18) is due to additivity and linearity of expectations. The linearity property is used to get to (19), after which elementary algebra leads to

the result.

$\square$

In the same fashion, it is possible to construct a proof for EU `(C2)` = EU `(C2)` + EU `(B2)`. However, as we know that EU `(B2)` = 0, this is trivial.

## A.2 MONTE CARLO APPROXIMATIONS

The measures we proposed through our framework, except for measure `(A1)`, incorporate posterior expectations $\mathrm{E}_{p(\boldsymbol{w}|\mathcal{D})}[\cdot]$. These are generally intractable to calculate exactly and are thus approximated through samples drawn from the distribution - a Monte Carlo approximation of the expectation. In this section we provide those approximations explicitly and discuss efficient ways to implement them, utilizing relationships between individual measures.

We assume that the posterior $p(\boldsymbol{w} \mid \mathcal{D})$ models to predict are drawn from and the posterior $p(\tilde{\boldsymbol{w}} \mid \mathcal{D})$ approximations of the true model are drawn from are the same. However, in practice it is generally the case that models for averaging are selected based on their accuracy on a validation set, or more generally that they are selected in a way optimal for predicting well. When sampling potential true models that are likely under the data, the functional diversity of samples is often of concern, e.g. as done with the sampling algorithm in Schweighofer et al. (2023b). This can be seen as either having different posteriors due to different priors or having different algorithms to obtain samples from the same posterior. However, for simplicity we state the MC approximations using only a single set of samples. The provided implementation however is able to handle also the case where different samples are used for the MC approximation.

**TU `(A2)`:**

$$\mathrm{CE}(p(\boldsymbol{y} \mid \boldsymbol{x}, \boldsymbol{w}) \; ; \; p(\boldsymbol{y} \mid \boldsymbol{x}, \mathcal{D})) \; = \; \mathrm{CE}(p(\boldsymbol{y} \mid \boldsymbol{x}, \boldsymbol{w}) \; ; \; \mathrm{E}_{\tilde{\boldsymbol{w}}}[p(\boldsymbol{y} \mid \boldsymbol{x}, \tilde{\boldsymbol{w}})]) \tag{22}$$

$$\approx \; \mathrm{CE}(p(\boldsymbol{y} \mid \boldsymbol{x}, \boldsymbol{w}) \; ; \; \frac{1}{M} \sum_{m=1}^{M} p(\boldsymbol{y} \mid \boldsymbol{x}, \tilde{\boldsymbol{w}}_m)), \qquad \tilde{\boldsymbol{w}}_m \sim p(\tilde{\boldsymbol{w}} \mid \mathcal{D})$$

**TU `(A3)`:**

$$\mathrm{E}_{\tilde{\boldsymbol{w}}}[\mathrm{CE}(p(\boldsymbol{y} \mid \boldsymbol{x}, \boldsymbol{w}) \; ; \; p(\boldsymbol{y} \mid \boldsymbol{x}, \tilde{\boldsymbol{w}}))] \tag{23}$$

$$\approx \; \frac{1}{M} \sum_{m=1}^{M} \mathrm{CE}(p(\boldsymbol{y} \mid \boldsymbol{x}, \boldsymbol{w}) \; ; \; p(\boldsymbol{y} \mid \boldsymbol{x}, \tilde{\boldsymbol{w}}_m)), \qquad \tilde{\boldsymbol{w}}_m \sim p(\tilde{\boldsymbol{w}} \mid \mathcal{D})$$

**TU `(B/C1)`:**

$$\mathrm{CE}(p(\boldsymbol{y} \mid \boldsymbol{x}, \mathcal{D}) \; ; \; p(\boldsymbol{y} \mid \boldsymbol{x}, \tilde{\boldsymbol{w}})) \; = \; \mathrm{E}_{\boldsymbol{w}}[\mathrm{CE}(p(\boldsymbol{y} \mid \boldsymbol{x}, \boldsymbol{w}) \; ; \; p(\boldsymbol{y} \mid \boldsymbol{x}, \tilde{\boldsymbol{w}}))] \tag{24}$$

$$\approx \; \frac{1}{N} \sum_{n=1}^{N} \mathrm{CE}(p(\boldsymbol{y} \mid \boldsymbol{x}, \boldsymbol{w}_n) \; ; \; p(\boldsymbol{y} \mid \boldsymbol{x}, \tilde{\boldsymbol{w}})), \qquad \boldsymbol{w}_n \sim p(\boldsymbol{w} \mid \mathcal{D})$$

**TU `(B/C2)`:**

$$\mathrm{CE}(p(\boldsymbol{y} \mid \boldsymbol{x}, \mathcal{D}) \; ; \; p(\boldsymbol{y} \mid \boldsymbol{x}, \mathcal{D})) \; = \; \mathrm{E}_{\boldsymbol{w}}[\mathrm{CE}(p(\boldsymbol{y} \mid \boldsymbol{x}, \boldsymbol{w}) \; ; \; p(\boldsymbol{y} \mid \boldsymbol{x}, \mathcal{D}))] \tag{25}$$

$$= \; \mathrm{H}(p(\boldsymbol{y} \mid \boldsymbol{x}, \mathcal{D})) \; = \; \mathrm{H}(\mathrm{E}_{\boldsymbol{w}}[p(\boldsymbol{y} \mid \boldsymbol{x}, \boldsymbol{w})])$$

$$\approx \; \mathrm{H}(\frac{1}{N} \sum_{n=1}^{N} p(\boldsymbol{y} \mid \boldsymbol{x}, \boldsymbol{w}_n)), \qquad \boldsymbol{w}_n \sim p(\boldsymbol{w} \mid \mathcal{D})$$

**TU `(B/C3)`:**

$$\mathrm{E}_{\tilde{\boldsymbol{w}}}[\mathrm{CE}((p(\boldsymbol{y} \mid \boldsymbol{x}, \mathcal{D}) \; ; \; p(\boldsymbol{y} \mid \boldsymbol{x}, \tilde{\boldsymbol{w}}))] \; = \; \mathrm{E}_{\boldsymbol{w}}[\mathrm{E}_{\tilde{\boldsymbol{w}}}[\mathrm{CE}(p(\boldsymbol{y} \mid \boldsymbol{x}, \boldsymbol{w}) \; ; \; p(\boldsymbol{y} \mid \boldsymbol{x}, \tilde{\boldsymbol{w}}))]] \tag{26}$$

$$\approx \; \frac{1}{NM} \sum_{n=1}^{N} \sum_{m=1}^{M} \mathrm{CE}(p(\boldsymbol{y} \mid \boldsymbol{x}, \boldsymbol{w}_n) \; ; \; p(\boldsymbol{y} \mid \boldsymbol{x}, \tilde{\boldsymbol{w}}_m)), \qquad \boldsymbol{w}_n \sim p(\boldsymbol{w} \mid \mathcal{D}), \; \tilde{\boldsymbol{w}}_m \sim p(\tilde{\boldsymbol{w}} \mid \mathcal{D})$$

**AU (B):**

$$H(p(\boldsymbol{y} \mid \boldsymbol{x}, \mathcal{D})) \;=\; H(E_{\boldsymbol{w}}\left[p(\boldsymbol{y} \mid \boldsymbol{x}, \boldsymbol{w})\right]) \tag{27}$$

$$\approx \; H(\frac{1}{N}\sum_{n=1}^{N} p(\boldsymbol{y} \mid \boldsymbol{x}, \boldsymbol{w}_n)), \qquad \boldsymbol{w}_n \sim p(\boldsymbol{w} \mid \mathcal{D})$$

**AU (C):**

$$E_{\boldsymbol{w}}\left[H(p(\boldsymbol{y} \mid \boldsymbol{x}, \boldsymbol{w}))\right] \approx \frac{1}{N}\sum_{n=1}^{N} H(p(\boldsymbol{y} \mid \boldsymbol{x}, \boldsymbol{w}_n)), \qquad \boldsymbol{w}_n \sim p(\boldsymbol{w} \mid \mathcal{D}) \tag{28}$$

**EU (A2):**

$$KL(p(\boldsymbol{y} \mid \boldsymbol{x}, \boldsymbol{w}) \parallel p(\boldsymbol{y} \mid \boldsymbol{x}, \mathcal{D})) \;=\; KL(p(\boldsymbol{y} \mid \boldsymbol{x}, \boldsymbol{w}) \parallel E_{\tilde{\boldsymbol{w}}}\left[p(\boldsymbol{y} \mid \boldsymbol{x}, \tilde{\boldsymbol{w}})\right]) \tag{29}$$

$$\approx \; KL(p(\boldsymbol{y} \mid \boldsymbol{x}, \boldsymbol{w}) \parallel \frac{1}{M}\sum_{m=1}^{M} p(\boldsymbol{y} \mid \boldsymbol{x}, \boldsymbol{w}_m)), \qquad \tilde{\boldsymbol{w}}_m \sim p(\tilde{\boldsymbol{w}} \mid \mathcal{D})$$

**EU (A3):**

$$E_{\tilde{\boldsymbol{w}}}\left[KL(p(\boldsymbol{y} \mid \boldsymbol{x}, \boldsymbol{w}) \parallel p(\boldsymbol{y} \mid \boldsymbol{x}, \tilde{\boldsymbol{w}}))\right] \tag{30}$$

$$\approx \; \frac{1}{M}\sum_{m=1}^{M} KL(p(\boldsymbol{y} \mid \boldsymbol{x}, \boldsymbol{w}) \parallel p(\boldsymbol{y} \mid \boldsymbol{x}, \boldsymbol{w}_m)), \qquad \tilde{\boldsymbol{w}}_m \sim p(\tilde{\boldsymbol{w}} \mid \mathcal{D})$$

**EU (B1):**

$$KL(p(\boldsymbol{y} \mid \boldsymbol{x}, \mathcal{D}) \parallel p(\boldsymbol{y} \mid \boldsymbol{x}, \tilde{\boldsymbol{w}})) \;=\; KL(E_{\boldsymbol{w}}\left[p(\boldsymbol{y} \mid \boldsymbol{x}, \boldsymbol{w})\right] \parallel p(\boldsymbol{y} \mid \boldsymbol{x}, \tilde{\boldsymbol{w}})) \tag{31}$$

$$\approx \; KL(\frac{1}{N}\sum_{n=1}^{N} p(\boldsymbol{y} \mid \boldsymbol{x}, \boldsymbol{w}_n) \parallel p(\boldsymbol{y} \mid \boldsymbol{x}, \tilde{\boldsymbol{w}})), \qquad \boldsymbol{w}_n \sim p(\boldsymbol{w} \mid \mathcal{D})$$

**EU (B3):**

$$E_{\tilde{\boldsymbol{w}}}\left[KL(p(\boldsymbol{y} \mid \boldsymbol{x}, \mathcal{D}) \parallel p(\boldsymbol{y} \mid \boldsymbol{x}, \tilde{\boldsymbol{w}}))\right] \;=\; E_{\tilde{\boldsymbol{w}}}\left[KL(E_{\boldsymbol{w}}\left[p(\boldsymbol{y} \mid \boldsymbol{x}, \boldsymbol{w})\right] \parallel p(\boldsymbol{y} \mid \boldsymbol{x}, \tilde{\boldsymbol{w}}))\right] \tag{32}$$

$$\approx \; \frac{1}{M}\sum_{m=1}^{M} KL(\frac{1}{N}\sum_{n=1}^{N} p(\boldsymbol{y} \mid \boldsymbol{x}, \boldsymbol{w}_n) \parallel p(\boldsymbol{y} \mid \boldsymbol{x}, \boldsymbol{w}_m)), \qquad \boldsymbol{w}_n \sim p(\boldsymbol{w} \mid \mathcal{D}),\; \tilde{\boldsymbol{w}}_m \sim p(\tilde{\boldsymbol{w}} \mid \mathcal{D})$$

**EU (C1):**

$$E_{\boldsymbol{w}}\left[KL(p(\boldsymbol{y} \mid \boldsymbol{x}, \boldsymbol{w}) \parallel p(\boldsymbol{y} \mid \boldsymbol{x}, \tilde{\boldsymbol{w}}))\right] \tag{33}$$

$$\approx \; \frac{1}{N}\sum_{n=1}^{N} KL(p(\boldsymbol{y} \mid \boldsymbol{x}, \boldsymbol{w}_n) \parallel p(\boldsymbol{y} \mid \boldsymbol{x}, \boldsymbol{w})), \qquad \boldsymbol{w}_n \sim p(\boldsymbol{w} \mid \mathcal{D})$$

**EU (C2):**

$$E_{\boldsymbol{w}}\left[KL(p(\boldsymbol{y} \mid \boldsymbol{x}, \boldsymbol{w}) \parallel p(\boldsymbol{y} \mid \boldsymbol{x}, \mathcal{D}))\right] \;=\; E_{\boldsymbol{w}}\left[KL(p(\boldsymbol{y} \mid \boldsymbol{x}, \boldsymbol{w}) \parallel E_{\tilde{\boldsymbol{w}}}\left[p(\boldsymbol{y} \mid \boldsymbol{x}, \tilde{\boldsymbol{w}})\right])\right] \tag{34}$$

$$\approx \; \frac{1}{N}\sum_{n=1}^{N} KL(p(\boldsymbol{y} \mid \boldsymbol{x}, \boldsymbol{w}_n) \parallel \frac{1}{M}\sum_{m=1}^{M} p(\boldsymbol{y} \mid \boldsymbol{x}, \boldsymbol{w}_m)), \qquad \boldsymbol{w}_n \sim p(\boldsymbol{w} \mid \mathcal{D}),\; \tilde{\boldsymbol{w}}_m \sim p(\tilde{\boldsymbol{w}} \mid \mathcal{D})$$

**EU (C3):**

$$E_{\boldsymbol{w}}\left[E_{\tilde{\boldsymbol{w}}}\left[KL(p(\boldsymbol{y} \mid \boldsymbol{x}, \boldsymbol{w}) \parallel p(\boldsymbol{y} \mid \boldsymbol{x}, \tilde{\boldsymbol{w}}))\right]\right] \tag{35}$$

$$\approx \; \frac{1}{NM}\sum_{n=1}^{N}\sum_{m=1}^{M} KL(p(\boldsymbol{y} \mid \boldsymbol{x}, \boldsymbol{w}_n) \parallel p(\boldsymbol{y} \mid \boldsymbol{x}, \boldsymbol{w}_m)), \qquad \boldsymbol{w}_n \sim p(\boldsymbol{w} \mid \mathcal{D}),\; \tilde{\boldsymbol{w}}_m \sim p(\tilde{\boldsymbol{w}} \mid \mathcal{D})$$

### A.3 GENERALIZATION TO RENYI CROSS-ENTROPY

In this section we review the Rényi cross-entropy which is a generalization of the cross-entropy discussed in the main paper. This allows to directly transfer our proposed measure of predictive uncertainty in Eq. (5) and the framework we introduced based on it (overview in Tab. 1) to other instances of Rényi cross-entropies.

Let us start with the Rényi entropy, which was proposed as a generalization of the Shannon entropy, in that for the limit of the Rényi parameter $\alpha \to 1$ the Rényi entropy becomes the Shannon entropy. For two discrete distributions $p$ and $q$ on the same support $\mathcal{Y}$ it is defined as

$$\mathrm{H}_\alpha(p) \;=\; \frac{1}{1-\alpha} \log \sum_i p_i^\alpha \tag{36}$$

Similarly, the Rényi divergence is a generalization of the Kullback-Leibler (KL) divergence, in that for the limit of the Rényi parameter $\alpha \to 1$ the Rényi divergence becomes the KL divergence. It is defined as

$$\mathrm{D}_\alpha(p \,||\, q) \;=\; \frac{1}{\alpha-1} \log \sum_i p_i^\alpha \, q_i^{1-\alpha} \tag{37}$$

Note that there are also versions of both for continuous distributions, basically exchanging the sum with an integral. However, the resulting Rényi differential entropy shares the same deficiencies as the Shannon differential entropy.

What is left is defining the Rényi cross-entropy. Motivated by the additive decomposition of Shannon cross-entropy into the entropy and KL divergence, Sarraf and Nie (2021) proposed to define the Rényi cross-entropy as

$$\mathrm{CE}_\alpha(p \,;\, q) \;:=\; \mathrm{H}_\alpha(p) \;+\; \mathrm{D}_\alpha(p \,||\, q) \tag{38}$$

Multiple closed form solutions for different values of $\alpha$ are already known for the Rényi entropy and divergence, making this a very simple solution. Furthermore, Valverde-Albacete and Peláez-Moreno (2019) introduced a closed form solution, which has been simplified to the following form by Thierrin et al. (2022):

$$\mathrm{CE}_\alpha(p \,;\, q) \;:=\; \frac{1}{1-\alpha} \log \sum_i p_i \, q_i^{\alpha-1} \tag{39}$$

Furthermore, Thierrin et al. (2022) proposes closed form solutions for this form of the Rényi differential cross-entropy for various continuous distributions.

In the following we stick to the definition of the Rényi cross-entropy by Sarraf and Nie (2021) (Eq. (38)) and state the respective entropy and divergence for special cases of $\alpha$. By defining the arbitrary discrete distributions as $p := p(y \mid \boldsymbol{x}, \cdot)$ and $q := p(y \mid \boldsymbol{x}, \boldsymbol{w}^*)$ each value of $\alpha$ yields a variant our proposed measure of predictive uncertainty (Eq. (5)), giving rise to variants of our proposed framework.

$\alpha = 0$:  The measure of entropy is called the Hartley or max-entropy, which is the cardinality of possible events $\mathcal{Y}$. It is given by

$$\mathrm{H}_0(p) := \log |\mathcal{Y}| \,. \tag{40}$$

The divergence is called max-divergence and is given by

$$\mathrm{D}_0(p \,||\, q) := -\log Q(\{i : p_i > 0\}) \,. \tag{41}$$

$\alpha = \frac{1}{2}$:  The measure of entropy is referred to as Bhattacharyya-entropy. It is given by

$$\mathrm{H}_{\frac{1}{2}}(p) := 2 \log \sum_i \sqrt{p_i} \,. \tag{42}$$

The divergence is called Bhattacharyya-divergence (minus twice the logarithm of the Bhattacharyya coefficient) and is given by

$$\mathrm{D}_{\frac{1}{2}}(p \,||\, q) := -2 \log \sum_i \sqrt{p_i q_i} \,. \tag{43}$$

$\alpha = 1$: This case is the well known Shannon-entropy, given by

$$\mathrm{H}_1(p) = \mathrm{H}(p) := - \sum_i p_i \log p_i .\tag{44}$$

The divergence is known as Kullback-Leibler divergence, given by

$$\mathrm{D}_1(p \parallel q) = \mathrm{KL}(p \parallel q) := \sum_i p_i \log \frac{p_i}{q_i} .\tag{45}$$

$\alpha = 2$: This case is called the collision entropy, which is closely related to the index of coincidence. It is given by

$$\mathrm{H}_2(p) := - \log \sum_i p_i^2 .\tag{46}$$

The corresponding divergence is based upon the chi-square divergence

$$\mathrm{D}_2(p \parallel q) := \log \left( \sum_{i=1}^{N} \frac{p_i^2}{q_i} \right) = \log \left( 1 + \sum_{i=1}^{N} \frac{(p_i - q_i)^2}{q_i} \right) .\tag{47}$$

$\alpha = \infty$: The entropy is known as the min-entropy. It is given by

$$\mathrm{H}_\infty(p) := - \log \max_i p_i .\tag{48}$$

The divergence

$$\mathrm{D}_\infty(p \parallel q) := \log \sup_i \frac{p_i}{q_i} .\tag{49}$$

**Notes.** Realizations of Renyi entropy satisfy the inequalities

$$\mathrm{H}_0(p) \geq \mathrm{H}_1(p) \geq \mathrm{H}_2(p) \geq \mathrm{H}_\infty(p)\tag{50}$$

Also Theorem 7 in van Erven and Harremos (2014) states that Renyi divergences are continuous in the order of $\alpha$.

### A.4 GENERALIZATION TO OTHER STRICTLY PROPER SCORING RULES

Another perspective on our measure of uncertainty (Eq. (5)) was recently proposed by Kotelevskii and Panov (2024) and Hofman et al. (2024b). They consider the zero-one, Brier, and Spherical score in addition to the log-score, which is the cross-entropy upon which the information-theoretic measures we discussed in the main paper are based (c.f. Eq. (5)). For the zero-one score, the resulting framework of measures is given in Tab. 2, for the Brier score in Tab. 3 and for the spherical score it is given in Tab. 4.

### A.5 ALTERNATIVE MEASURE

The reverse order of the arguments for the cross-entropy in Eq. (5), that is, $\mathrm{CE}(p(y \mid \boldsymbol{x}, \boldsymbol{w}^*)\ ;\ p(y \mid \boldsymbol{x}, \cdot))$, gives rise to an alternative measure that is consistent with Eq. (1). This measure, also known as "pointwise risk" under the log score at an input (point) $\boldsymbol{x}$, has been considered as a measure of predictive uncertainty (Gruber and Buettner, 2023; Lahlou et al., 2023; Kotelevskii and Panov, 2024; Hofman et al., 2024b). However, we argue that our proposed measure (Eq. (5)) is more meaningful. Our measure considers the uncertainty inherent to predicting with the selected model, plus the uncertainty due to any potential mismatch with the true model. The alternative measure considers the uncertainty inherent to predicting with the true model, plus the uncertainty due to any potential mismatch with the selected model. However, we generally don't know the true model, thus can't actually use it to predict and have to resort to an approximation of the true model anyways.

Table 2: **Our proposed framework applied under the zero-one score.** Each measure denotes a different instantiation of our proposed measure given by Eq. (5), but using the zero-one score instead of the cross-entropy (log score) for different assumptions about the predicting model and how the true model is approximated. For brevity, we define $p_{\boldsymbol{w}} \coloneqq p(y \mid \boldsymbol{x}, \boldsymbol{w})$, $p_{\mathcal{D}} \coloneqq p(y \mid \boldsymbol{x}, \mathcal{D})$, $\mathrm{E}_{\boldsymbol{w}} \coloneqq \mathrm{E}_{p(\boldsymbol{w}|\mathcal{D})}$ (the same for $\tilde{\boldsymbol{w}}$) and $p_{\bullet}(\widehat{p_\circ}) \coloneqq p(y = \arg\max p(y \mid \boldsymbol{x}, \circ) \mid \boldsymbol{x}, \bullet)$. Expressions with the same cell coloring are equivalent to each other. Each measure of total uncertainty additively decomposes into an aleatoric and epistemic component.

| Predicting model | | Approximation of true predictive distribution | | |
|---|---|---|---|---|
| | | (1) $\tilde{\boldsymbol{w}}$ | (2) $\mathrm{E}_{\tilde{\boldsymbol{w}}}$ | (3) $\tilde{\boldsymbol{w}} \sim p(\tilde{\boldsymbol{w}} \mid \mathcal{D})$ |
| TU | (A) $\boldsymbol{w}$ | $1 - p_{\boldsymbol{w}}(\widehat{p_{\tilde{\boldsymbol{w}}}})$ | $1 - p_{\boldsymbol{w}}(\widehat{p_{\mathcal{D}}})$ | $\mathrm{E}_{\tilde{\boldsymbol{w}}}\left[1 - p_{\boldsymbol{w}}(\widehat{p_{\tilde{\boldsymbol{w}}}})\right]$ |
| | (B) $\mathrm{E}_{\boldsymbol{w}}$ | $1 - p_{\mathcal{D}}(\widehat{p_{\tilde{\boldsymbol{w}}}})$ | $1 - p_{\mathcal{D}}(\widehat{p_{\mathcal{D}}})$ | $\mathrm{E}_{\tilde{\boldsymbol{w}}}\left[1 - p_{\mathcal{D}}(\widehat{p_{\tilde{\boldsymbol{w}}}})\right]$ |
| | (C) $\boldsymbol{w} \sim p(\boldsymbol{w} \mid \mathcal{D})$ | $\mathrm{E}_{\boldsymbol{w}}\left[1 - p_{\boldsymbol{w}}(\widehat{p_{\tilde{\boldsymbol{w}}}})\right]$ | $\mathrm{E}_{\boldsymbol{w}}\left[1 - p_{\boldsymbol{w}}(\widehat{p_{\mathcal{D}}})\right]$ | $\mathrm{E}_{\boldsymbol{w}}\left[\mathrm{E}_{\tilde{\boldsymbol{w}}}\left[1 - p_{\boldsymbol{w}}(\widehat{p_{\tilde{\boldsymbol{w}}}})\right]\right]$ |
| AU | (A) $\boldsymbol{w}$ | $1 - \max p_{\boldsymbol{w}}$ | $1 - \max p_{\boldsymbol{w}}$ | $1 - \max p_{\boldsymbol{w}}$ |
| | (B) $\mathrm{E}_{\boldsymbol{w}}$ | $1 - \max p_{\mathcal{D}}$ | $1 - \max p_{\mathcal{D}}$ | $1 - \max p_{\mathcal{D}}$ |
| | (C) $\boldsymbol{w} \sim p(\boldsymbol{w} \mid \mathcal{D})$ | $1 - \mathrm{E}_{\boldsymbol{w}}\left[\max p_{\boldsymbol{w}}\right]$ | $1 - \mathrm{E}_{\boldsymbol{w}}\left[\max p_{\boldsymbol{w}}\right]$ | $1 - \mathrm{E}_{\boldsymbol{w}}\left[\max p_{\boldsymbol{w}}\right]$ |
| EU | (A) $\boldsymbol{w}$ | $\max p_{\boldsymbol{w}} - p_{\boldsymbol{w}}(\widehat{p_{\tilde{\boldsymbol{w}}}})$ | $\max p_{\boldsymbol{w}} - p_{\boldsymbol{w}}(\widehat{p_{\mathcal{D}}})$ | $\mathrm{E}_{\tilde{\boldsymbol{w}}}\left[\max p_{\boldsymbol{w}} - p_{\boldsymbol{w}}(\widehat{p_{\tilde{\boldsymbol{w}}}})\right]$ |
| | (B) $\mathrm{E}_{\boldsymbol{w}}$ | $\max p_{\mathcal{D}} - p_{\mathcal{D}}(\widehat{p_{\tilde{\boldsymbol{w}}}})$ | $\max p_{\mathcal{D}} \overset{0}{\nearrow} p_{\mathcal{D}}(\widehat{p_{\mathcal{D}}})$ | $\mathrm{E}_{\tilde{\boldsymbol{w}}}\left[\max p_{\mathcal{D}} - p_{\mathcal{D}}(\widehat{p_{\tilde{\boldsymbol{w}}}})\right]$ |
| | (C) $\boldsymbol{w} \sim p(\boldsymbol{w} \mid \mathcal{D})$ | $\mathrm{E}_{\boldsymbol{w}}\left[\max p_{\boldsymbol{w}} - p_{\boldsymbol{w}}(\widehat{p_{\tilde{\boldsymbol{w}}}})\right]$ | $\mathrm{E}_{\boldsymbol{w}}\left[\max p_{\boldsymbol{w}} - p_{\boldsymbol{w}}(\widehat{p_{\mathcal{D}}})\right]$ | $\mathrm{E}_{\boldsymbol{w}}\left[\mathrm{E}_{\tilde{\boldsymbol{w}}}\left[\max p_{\boldsymbol{w}} - p_{\boldsymbol{w}}(\widehat{p_{\tilde{\boldsymbol{w}}}})\right]\right]$ |

Table 3: **Our proposed framework applied under the Brier score.** Each measure denotes a different instantiation of our proposed measure given by Eq. (5), but using the Brier score instead of the cross-entropy (log score) for different assumptions about the predicting model and how the true model is approximated. For brevity, we define $p_{\boldsymbol{w}} \coloneqq p(y \mid \boldsymbol{x}, \boldsymbol{w})$, $p_{\mathcal{D}} \coloneqq p(y \mid \boldsymbol{x}, \mathcal{D})$, and $\mathrm{E}_{\boldsymbol{w}} \coloneqq \mathrm{E}_{p(\boldsymbol{w}|\mathcal{D})}$ (the same for $\tilde{\boldsymbol{w}}$). The 2-norm is defined as $\|p(y \mid \boldsymbol{x}, \bullet)\|_2 \coloneqq \sqrt{\sum_{k=1}^{K} p(y = k \mid \boldsymbol{x}, \bullet)^2}$. Expressions with the same cell coloring are equivalent to each other. Each measure of total uncertainty additively decomposes into an aleatoric and epistemic component.

| Predicting model | | Approximation of true predictive distribution | | |
|---|---|---|---|---|
| | | (1) $\tilde{\boldsymbol{w}}$ | (2) $\mathrm{E}_{\tilde{\boldsymbol{w}}}$ | (3) $\tilde{\boldsymbol{w}} \sim p(\tilde{\boldsymbol{w}} \mid \mathcal{D})$ |
| TU | (A) $\boldsymbol{w}$ | $1 - \|p_{\boldsymbol{w}}\|_2^2 + \|p_{\boldsymbol{w}} - p_{\tilde{\boldsymbol{w}}}\|_2^2$ | $1 - \|p_{\boldsymbol{w}}\|_2^2 + \|p_{\boldsymbol{w}} - p_{\mathcal{D}}\|_2^2$ | $\mathrm{E}_{\tilde{\boldsymbol{w}}}\left[1 - \|p_{\boldsymbol{w}}\|_2^2 + \|p_{\boldsymbol{w}} - p_{\tilde{\boldsymbol{w}}}\|_2^2\right]$ |
| | (B) $\mathrm{E}_{\boldsymbol{w}}$ | $1 - \|p_{\mathcal{D}}\|_2^2 + \|p_{\mathcal{D}} - p_{\tilde{\boldsymbol{w}}}\|_2^2$ | $1 - \|p_{\mathcal{D}}\|_2^2 + \|p_{\mathcal{D}} \overset{0}{\nearrow} p_{\mathcal{D}}\|_2^2$ | $\mathrm{E}_{\tilde{\boldsymbol{w}}}\left[1 - \|p_{\mathcal{D}}\|_2^2 + \|p_{\mathcal{D}} - p_{\tilde{\boldsymbol{w}}}\|_2^2\right]$ |
| | (C) $\boldsymbol{w} \sim p(\boldsymbol{w} \mid \mathcal{D})$ | $\mathrm{E}_{\boldsymbol{w}}\left[1 - \|p_{\boldsymbol{w}}\|_2^2 + \|p_{\boldsymbol{w}} - p_{\tilde{\boldsymbol{w}}}\|_2^2\right]$ | $\mathrm{E}_{\boldsymbol{w}}\left[1 - \|p_{\boldsymbol{w}}\|_2^2 + \|p_{\boldsymbol{w}} - p_{\mathcal{D}}\|_2^2\right]$ | $\mathrm{E}_{\boldsymbol{w}}\left[\mathrm{E}_{\tilde{\boldsymbol{w}}}\left[1 - \|p_{\boldsymbol{w}}\|_2^2 + \|p_{\boldsymbol{w}} - p_{\tilde{\boldsymbol{w}}}\|_2^2\right]\right]$ |
| AU | (A) $\boldsymbol{w}$ | $1 - \|p_{\boldsymbol{w}}\|_2^2$ | $1 - \|p_{\boldsymbol{w}}\|_2^2$ | $1 - \|p_{\boldsymbol{w}}\|_2^2$ |
| | (B) $\mathrm{E}_{\boldsymbol{w}}$ | $1 - \|p_{\mathcal{D}}\|_2^2$ | $1 - \|p_{\mathcal{D}}\|_2^2$ | $1 - \|p_{\mathcal{D}}\|_2^2$ |
| | (C) $\boldsymbol{w} \sim p(\boldsymbol{w} \mid \mathcal{D})$ | $\mathrm{E}_{\boldsymbol{w}}\left[1 - \|p_{\boldsymbol{w}}\|_2^2\right]$ | $\mathrm{E}_{\boldsymbol{w}}\left[1 - \|p_{\boldsymbol{w}}\|_2^2\right]$ | $\mathrm{E}_{\boldsymbol{w}}\left[1 - \|p_{\boldsymbol{w}}\|_2^2\right]$ |
| EU | (A) $\boldsymbol{w}$ | $\|p_{\boldsymbol{w}} - p_{\tilde{\boldsymbol{w}}}\|_2^2$ | $\|p_{\boldsymbol{w}} - p_{\mathcal{D}}\|_2^2$ | $\mathrm{E}_{\tilde{\boldsymbol{w}}}\left[\|p_{\boldsymbol{w}} - p_{\tilde{\boldsymbol{w}}}\|_2^2\right]$ |
| | (B) $\mathrm{E}_{\boldsymbol{w}}$ | $\|p_{\mathcal{D}} - p_{\tilde{\boldsymbol{w}}}\|_2^2$ | $\|p_{\mathcal{D}} \overset{0}{\nearrow} p_{\mathcal{D}}\|_2^2$ | $\mathrm{E}_{\tilde{\boldsymbol{w}}}\left[\|p_{\mathcal{D}} - p_{\tilde{\boldsymbol{w}}}\|_2^2\right]$ |
| | (C) $\boldsymbol{w} \sim p(\boldsymbol{w} \mid \mathcal{D})$ | $\mathrm{E}_{\boldsymbol{w}}\left[\|p_{\boldsymbol{w}} - p_{\tilde{\boldsymbol{w}}}\|_2^2\right]$ | $\mathrm{E}_{\boldsymbol{w}}\left[\|p_{\boldsymbol{w}} - p_{\mathcal{D}}\|_2^2\right]$ | $\mathrm{E}_{\boldsymbol{w}}\left[\mathrm{E}_{\tilde{\boldsymbol{w}}}\left[\|p_{\boldsymbol{w}} - p_{\tilde{\boldsymbol{w}}}\|_2^2\right]\right]$ |

Table 4: **Our proposed framework applied under the spherical score.** Each measure denotes a different instantiation of our proposed measure given by Eq. (5), but using the spherical score instead of the cross-entropy (log score) for different assumptions about the predicting model and how the true model is approximated. For brevity, we define $p_{\boldsymbol{w}} := p(y \mid \boldsymbol{x}, \boldsymbol{w})$, $p_{\mathcal{D}} := p(y \mid \boldsymbol{x}, \mathcal{D})$, and $\mathrm{E}_{\boldsymbol{w}} := \mathrm{E}_{p(\boldsymbol{w}\mid\mathcal{D})}$ (the same for $\tilde{\boldsymbol{w}}$). The 2-norm is defined as $\|p_{\bullet}\|_2 := \sqrt{\sum_{k=1}^{K} p(y = k \mid \boldsymbol{x}, \bullet)^2}$. Furthermore, the scalar product is defined as $\langle p_{\bullet}, p_{\circ}\rangle := \sum_{k=1}^{K} p(y = k \mid \boldsymbol{x}, \bullet) \cdot p(y = k \mid \boldsymbol{x}, \circ)$. Expressions with the same cell coloring are equivalent to each other. Each measure of total uncertainty additively decomposes into an aleatoric and epistemic component.

| | Predicting model | Approximation of true predictive distribution | | |
| --- | --- | --- | --- | --- |
| | | (1) $\tilde{\boldsymbol{w}}$ | (2) $\mathrm{E}_{\tilde{\boldsymbol{w}}}$ | (3) $\tilde{\boldsymbol{w}} \sim p(\tilde{\boldsymbol{w}} \mid \mathcal{D})$ |
| **TU** | (A) $\boldsymbol{w}$ | $1 - \frac{\langle p_w, p_{\tilde w}\rangle}{\|p_{\tilde w}\|_2}$ | $1 - \frac{\langle p_w, p_{\mathcal D}\rangle}{\|p_{\mathcal D}\|_2}$ | $\mathrm{E}_{\tilde w}\left[1 - \frac{\langle p_w, p_{\tilde w}\rangle}{\|p_{\tilde w}\|_2}\right]$ |
| | (B) $\mathrm{E}_{\boldsymbol{w}}$ | $1 - \frac{\langle p_{\mathcal D}, p_{\tilde w}\rangle}{\|p_{\tilde w}\|_2}$ | $1 - \frac{\langle p_{\mathcal D}, p_{\mathcal D}\rangle}{\|p_{\mathcal D}\|_2}$ | $\mathrm{E}_{\tilde w}\left[1 - \frac{\langle p_{\mathcal D}, p_{\tilde w}\rangle}{\|p_{\tilde w}\|_2}\right]$ |
| | (C) $\boldsymbol{w} \sim p(\boldsymbol{w} \mid \mathcal{D})$ | $\mathrm{E}_{w}\left[1 - \frac{\langle p_w, p_{\tilde w}\rangle}{\|p_{\tilde w}\|_2}\right]$ | $\mathrm{E}_{w}\left[1 - \frac{\langle p_w, p_{\mathcal D}\rangle}{\|p_{\mathcal D}\|_2}\right]$ | $\mathrm{E}_{w}\left[\mathrm{E}_{\tilde w}\left[1 - \frac{\langle p_w, p_{\tilde w}\rangle}{\|p_{\tilde w}\|_2}\right]\right]$ |
| **AU** | (A) $\boldsymbol{w}$ | $1 - \|p_w\|_2$ | $1 - \|p_w\|_2$ | $1 - \|p_w\|_2$ |
| | (B) $\mathrm{E}_{\boldsymbol{w}}$ | $1 - \|p_{\mathcal D}\|_2$ | $1 - \|p_{\mathcal D}\|_2$ | $1 - \|p_{\mathcal D}\|_2$ |
| | (C) $\boldsymbol{w} \sim p(\boldsymbol{w} \mid \mathcal{D})$ | $\mathrm{E}_{w}\left[1 - \|p_w\|_2\right]$ | $\mathrm{E}_{w}\left[1 - \|p_w\|_2\right]$ | $\mathrm{E}_{w}\left[1 - \|p_w\|_2\right]$ |
| **EU** | (A) $\boldsymbol{w}$ | $\|p_w\|_2 - \frac{\langle p_w, p_{\tilde w}\rangle}{\|p_{\tilde w}\|_2}$ | $\|p_w\|_2 - \frac{\langle p_w, p_{\mathcal D}\rangle}{\|p_{\mathcal D}\|_2}$ | $\mathrm{E}_{\tilde w}\left[\|p_w\|_2 - \frac{\langle p_w, p_{\tilde w}\rangle}{\|p_{\tilde w}\|_2}\right]$ |
| | (B) $\mathrm{E}_{\boldsymbol{w}}$ | $\|p_{\mathcal D}\|_2 - \frac{\langle p_{\mathcal D}, p_{\tilde w}\rangle}{\|p_{\tilde w}\|_2}$ | $\|p_{\mathcal D}\|_2 - \frac{\langle p_{\mathcal D}, p_{\mathcal D}\rangle}{\|p_{\mathcal D}\|_2} \nearrow^{0}$ | $\mathrm{E}_{\tilde w}\left[\|p_{\mathcal D}\|_2 - \frac{\langle p_{\mathcal D}, p_{\tilde w}\rangle}{\|p_{\tilde w}\|_2}\right]$ |
| | (C) $\boldsymbol{w} \sim p(\boldsymbol{w} \mid \mathcal{D})$ | $\mathrm{E}_{w}\left[\|p_w\|_2 - \frac{\langle p_w, p_{\tilde w}\rangle}{\|p_{\tilde w}\|_2}\right]$ | $\mathrm{E}_{w}\left[\|p_w\|_2 - \frac{\langle p_w, p_{\mathcal D}\rangle}{\|p_{\mathcal D}\|_2}\right]$ | $\mathrm{E}_{w}\left[\mathrm{E}_{\tilde w}\left[\|p_w\|_2 - \frac{\langle p_w, p_{\tilde w}\rangle}{\|p_{\tilde w}\|_2}\right]\right]$ |

## A.6 REGRESSION

For a probabilistic regression model, e.g. under a Gaussian assumption, the distribution parameters are estimated, i.e. mean and variance for the Gaussian predictive distribution. The model is then trained by minimizing the negative log-likelihood under the training dataset.

Many works follow Depeweg et al. (2018) and utilize a variance decomposition for uncertainty quantification, where the aleatoric component is the expected variance and the epistemic component is the variance of means, where expectation and variance are over the model posterior. However, Depeweg et al. (2018) also consider the uncertainty measure given by Eq. (3), using differential entropies for the continuous predictive distributions. The same can be done in order to adapt our framework in Tab. 1 for continuous predictive distributions.

Nevertheless, there are two important drawbacks one need to consider when doing this. First, differential entropy can be unbounded, depending on the nature of the predictive distribution. For the example of a Gaussian, it can be between $-\infty$ and $\infty$. In addition, it is not invariant to a change of variables, making it a relative rather than an absolute measure. Second, the posterior predictive distribution as defined in Eq. (2) is generally a mixture of individual distributions, unlike in the discrete case. This makes MC approximations of the resulting measures more involved.

# B ADDITIONAL EXPERIMENTS

In this section, we provide additional empirical results of our evaluation of the proposed framework of uncertainty measures.
The code to reproduce our experiments will be made public upon acceptance.

## B.1 ILLUSTRATIVE EXAMPLE

Here, we provide an illustrative synthetic example often discussed in the literature (Wimmer et al., 2023; Schweighofer et al., 2023a; Sale et al., 2023b). We consider a predictor defined as a Bernoulli distribution leading to the predictive distribution $p(y \mid \theta)$. Thus, there is no model involved for mapping from the input space to the Bernoulli parameter. The only free parameter is the Bernoulli parameter. Therefore, the posterior distribution is defined as $p(\theta \mid \mathcal{D}) = p(\mathcal{D} \mid \theta)p(\theta)/p(\mathcal{D})$. To examplify our framework, we consider a Beta posterior distribution $Beta(\theta; 2, 3)$. The true Bernoulli parameter $\theta^*$ is not known.

Results are shown in Fig. 6, depicting what is considered as predicting model (green) and what is compared to as approximation of the true model. The green line for measures (A1/2/3) and the violet line for measures (A/B/C1) were chosen arbitrarily, but different to the expected Bernoulli parameter to exemplify the differences between measures.

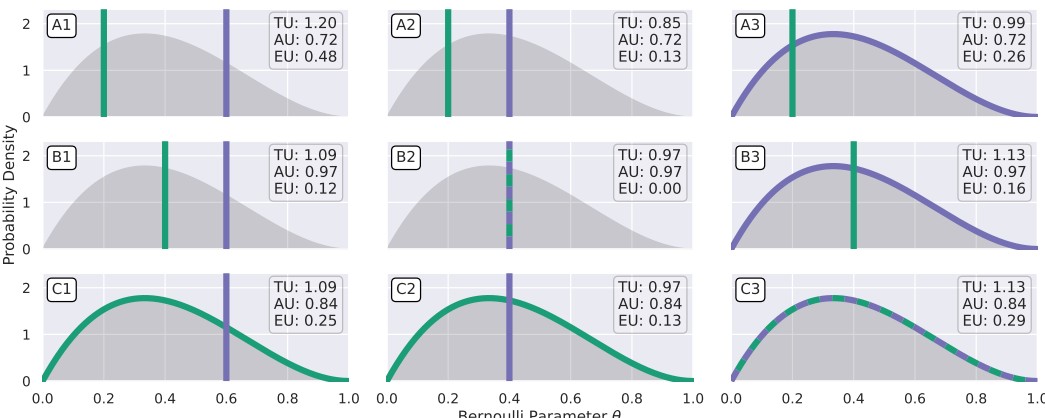

Figure 6: Uncertainty measures given by the predicting model and the approximation of the true model. We consider the posterior distribution $Beta(\theta; 2, 3)$, shaded in gray.

## B.2 SELECTIVE PREDICTION

We provide the additional results for selective prediction as discussed in the main paper.

The results for predicting under a single model are shown in Fig. 7. We observe, that the best measure for DE and LA is TU (A3), as well as TU (A2) in the case of MCD. Overall, measures that consider the single model as predicting model perform well throughout comparing within TU, AU and EU. Again, EU (A2) performs surprisingly bad for LA as posterior sampling method. For the local methods LA and MCD, AU (A) is better than AU (B) and AU (C), which is not the case for DE.

The additional results for predicting under the average model with MCD are shown in Fig. 8. Overall, the results are very similar to the other local posterior sampling method LA provided in Fig. 4. However, TU (A2) is the best measure for MCD, while it is TU (A3) for LA. For the global posterior sampling method DE however, TU (B/C3) performs best.

Finally, the results for predicting under a model according to the posterior are given in Fig. 9. The results are very similar to the results under the average model in Fig. 4 and Fig. 8. However, the difference between the different AU measures for LA and MCD is extremely tight. This is also the case for the TU measures, yet to a lesser extent.

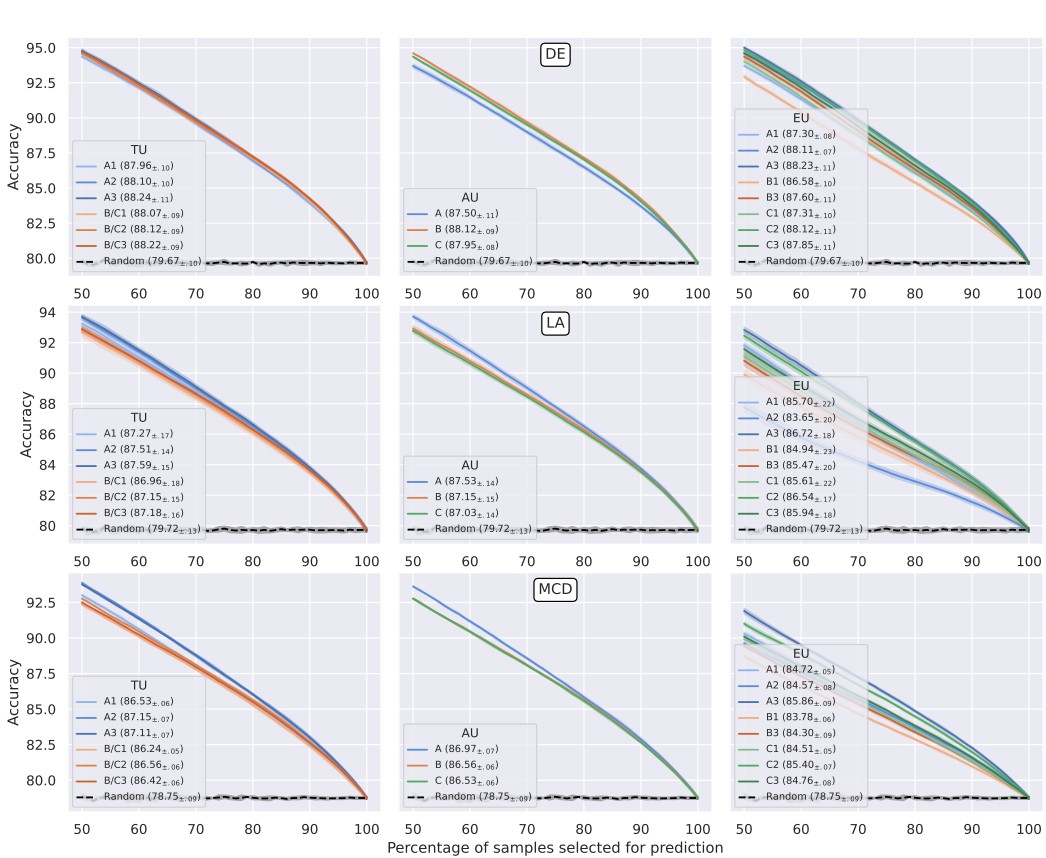

Figure 7: **Selective prediction results under a single model.** Accuracies per fraction of datapoints a single model predicts on, as well as area under the accuracy retention curve (tabulated in legend) using different proposed measures of uncertainty as score. Uncertainty measures are approximated by DE (top row), LA (middle row) and MCD (bottom row) as posterior sampling method. Accuracies are averaged over all datasets. Means and standard deviations are calculated using five independent runs.

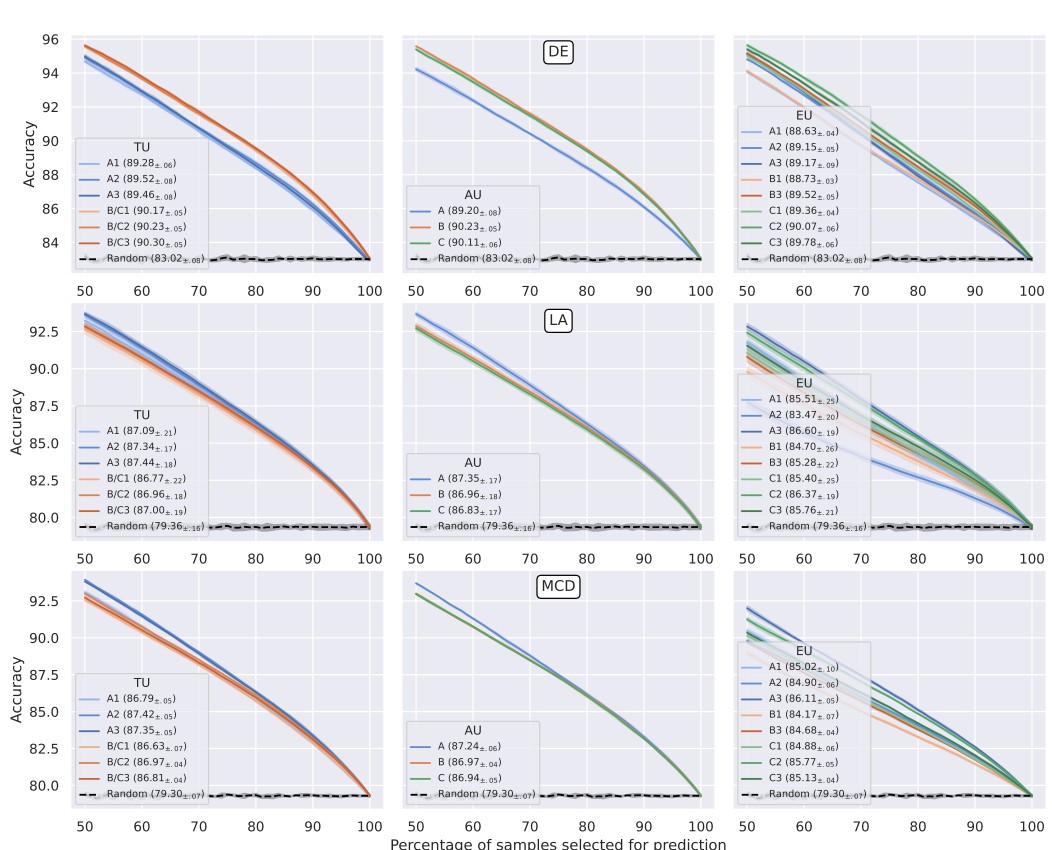

Figure 8: **Selective prediction results under the average model.** Accuracies per fraction of datapoints the average model predicts on, as well as area under the accuracy retention curve (tabulated in legend) using different proposed measures of uncertainty as score. Uncertainty measures are approximated by DE (top row), LA (middle row) and MCD (bottom row) as posterior sampling method. Accuracies are averaged over all datasets. Means and standard deviations are calculated using five independent runs.

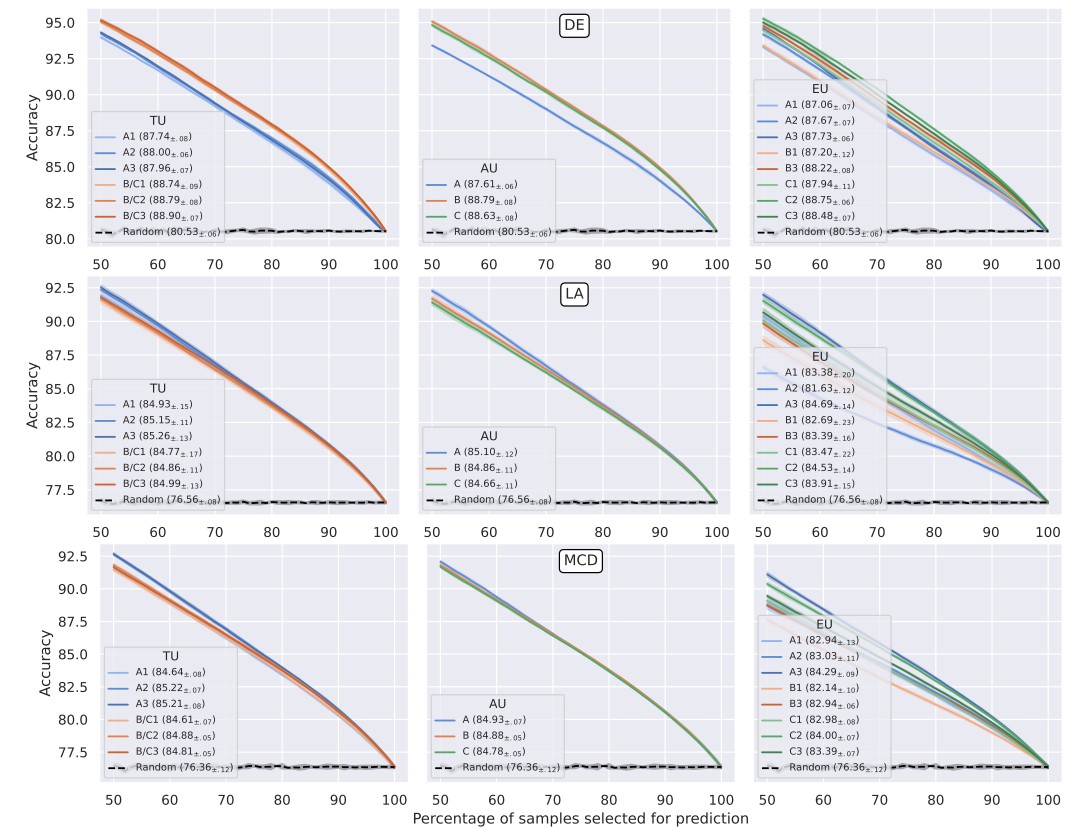

Figure 9: **Selective prediction results under a model according to the posterior.** Accuracies per fraction of datapoints a model drawn according to the posterior predicts on, as well as area under the accuracy retention curve (tabulated in legend) using different proposed measures of uncertainty as score. Uncertainty measures are approximated by DE (top row), LA (middle row) and MCD (bottom row) as posterior sampling method. Accuracies are averaged over all datasets. Means and standard deviations are calculated using five independent runs.

### B.3 DETAILED RESULTS

The results for misclassification detection and OOD detection in the main paper show aggregate performances over multiple datasets to provide more robust conclusions about the performance of individual measures of uncertainty. In this section we provide individual results for completeness.

**Misclassification detection.** The detailed results for misclassification detection are given in Fig. 10 for a single predicting model, in Fig. 11 for the average predicting model as well as in Fig. 12 for predicting with a model according to the posterior. Although there are nuanced differences between datasets, conclusions translate very well between them for a given posterior sampling method.

**OOD detection** The detailed results for OOD detection for CIFAR10 as ID dataset are given in Fig. 13, for CIFAR100 as ID dataset in Fig. 14, for SVHN as ID dataset in Fig. 15 and for TIN as ID dataset in Fig. 16. We observe the highest variability of experiments for TIN as ID dataset, where there is high variability for both the OOD dataset as well as for the posterior sampling method used. For the other ID datasets, the main variability comes from the posterior sampling methods and different OOD datasets lead to very similar results.

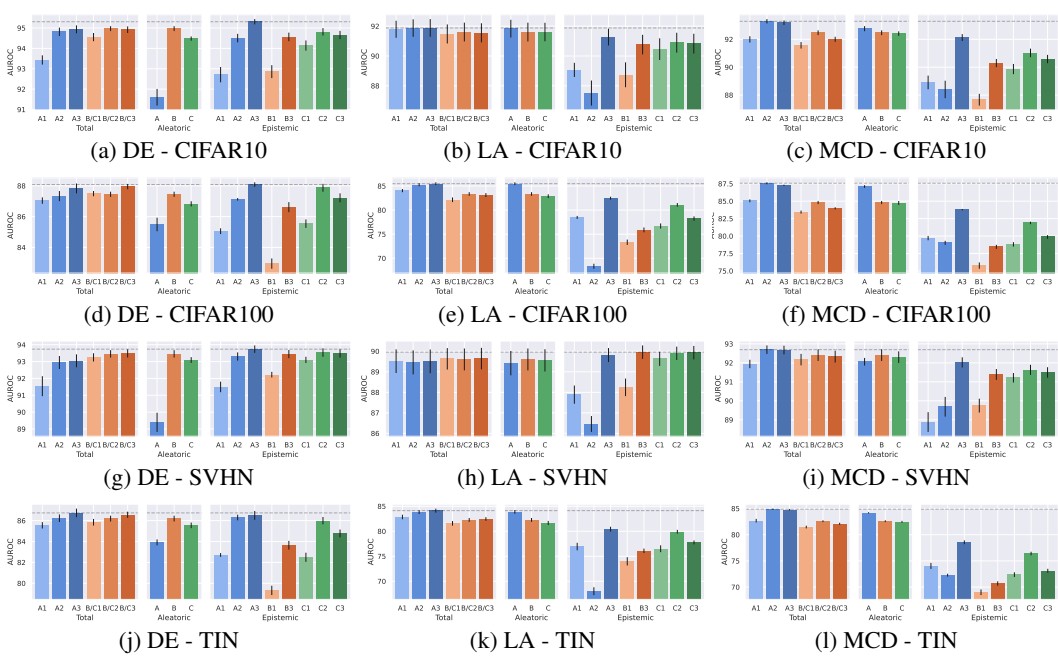

Figure 10: **Misclassification detection results under single predicting model.** AUROC for distinguishing between correctly and incorrectly predicted datapoints under a single predicting model, using the different proposed measures of uncertainty as score. Means and standard deviations are calculated using five independent runs.

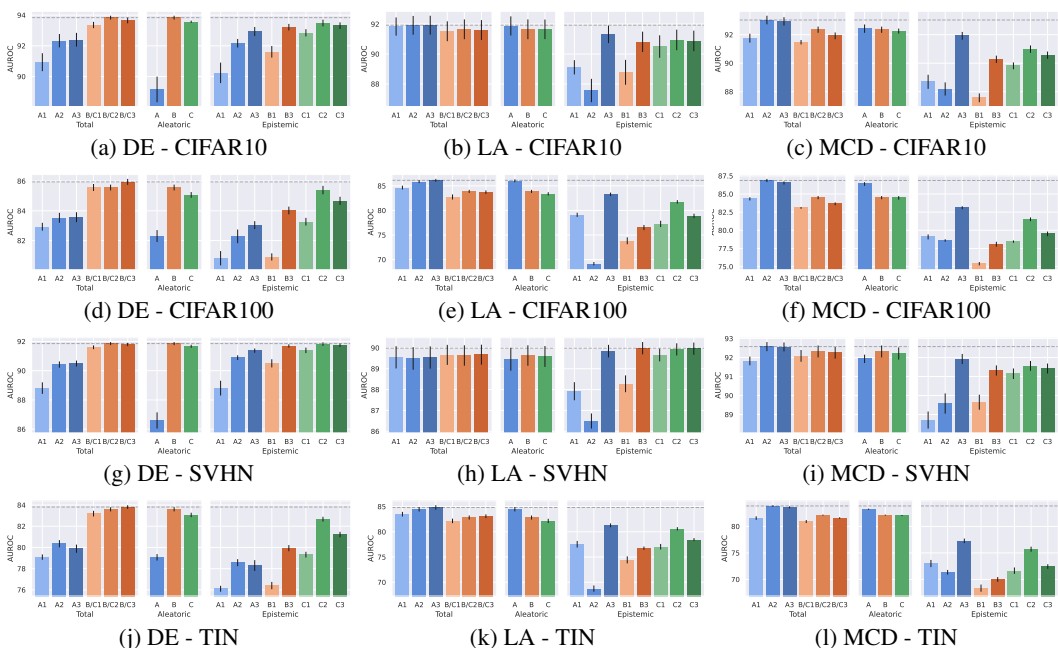

Figure 11: **Misclassification detection results under average predicting model.** AUROC for distinguishing between correctly and incorrectly predicted datapoints under the average predicting model, using the different proposed measures of uncertainty as score. Means and standard deviations are calculated using five independent runs.

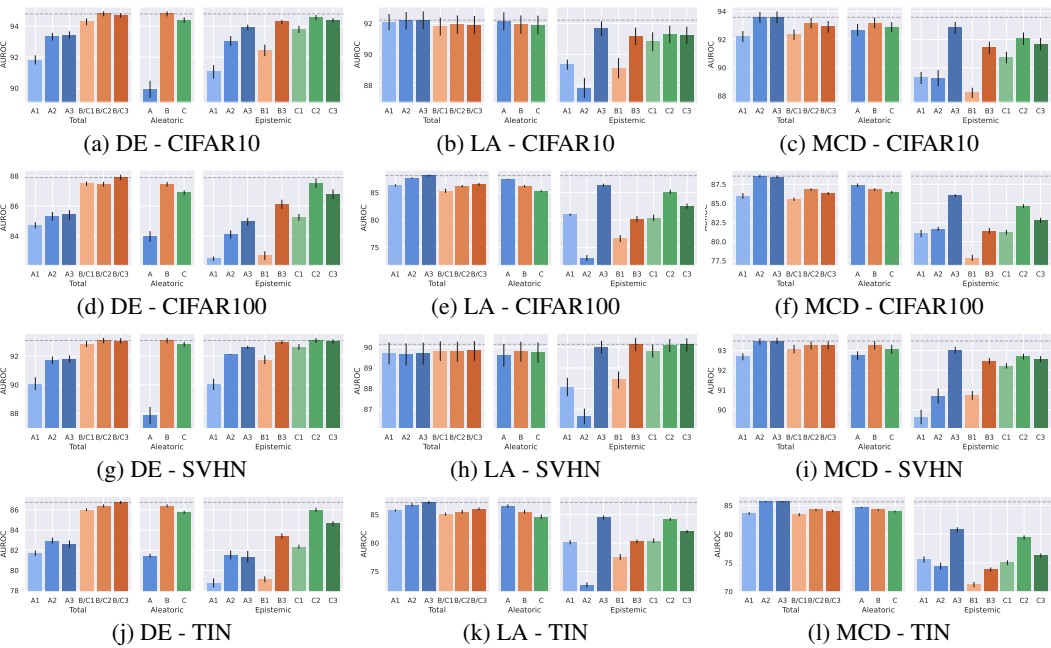

Figure 12: **Misclassification detection results under model according to posterior predicting.** AUROC for distinguishing between correctly and incorrectly predicted datapoints under a model according to posterior predicting, using the different proposed measures of uncertainty as score. Means and standard deviations are calculated using five independent runs.

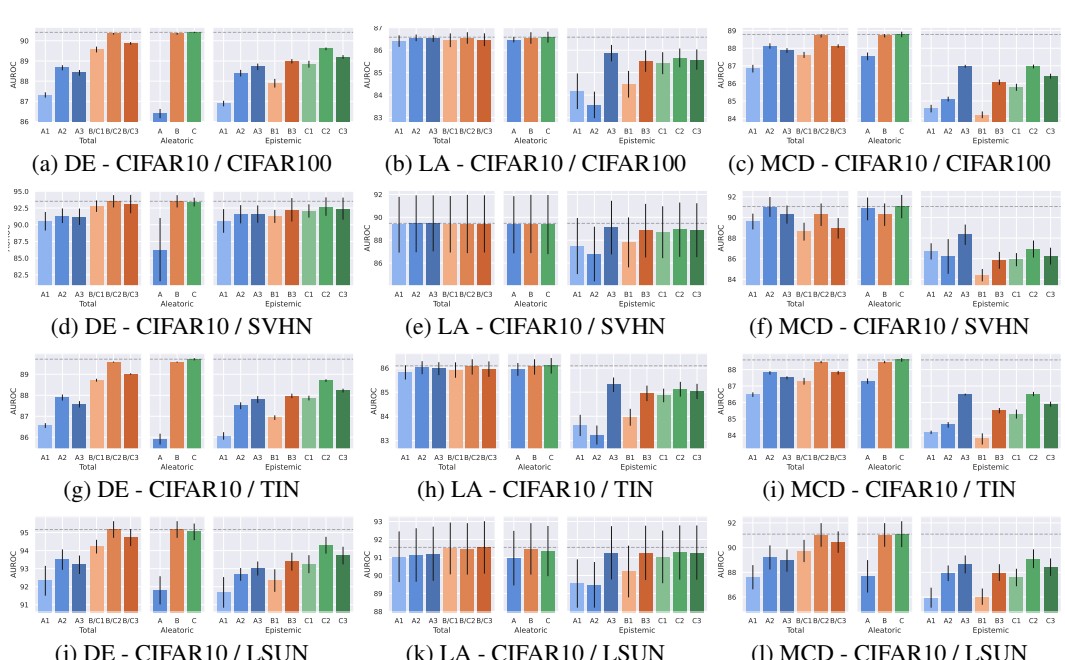

Figure 13: **OOD detection results for CIFAR10.** AUROC for distinguishing between ID and OOD datapoints using the different proposed measures of uncertainty as score. Means and standard deviations are calculated using five independent runs.

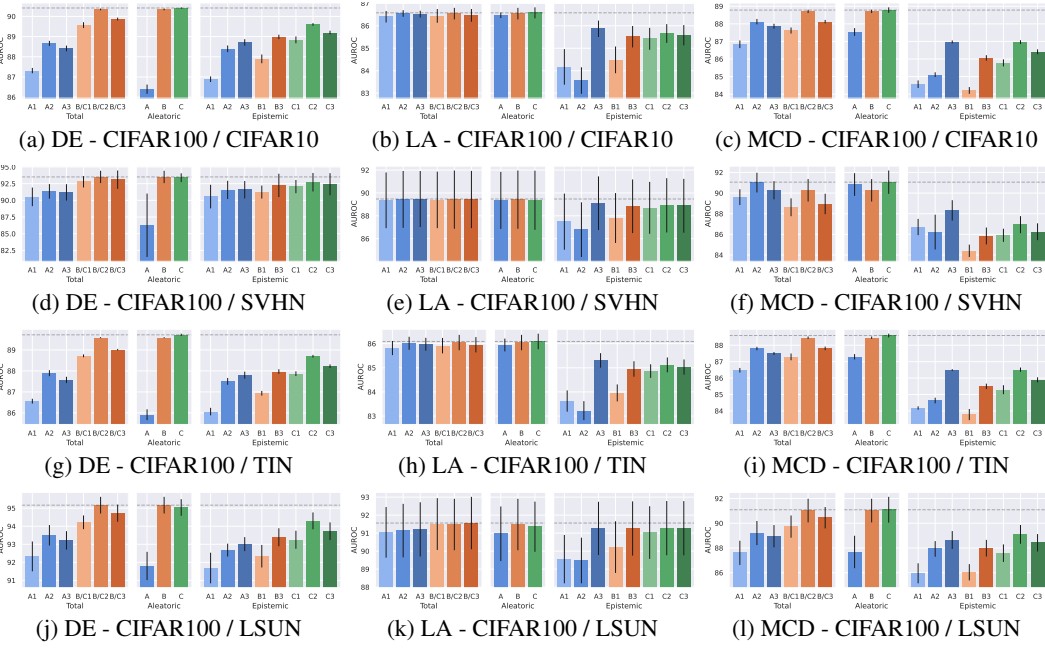

Figure 14: **OOD detection results for CIFAR100.** AUROC for distinguishing between ID and OOD datapoints using the different proposed measures of uncertainty as score. Means and standard deviations are calculated using five independent runs.

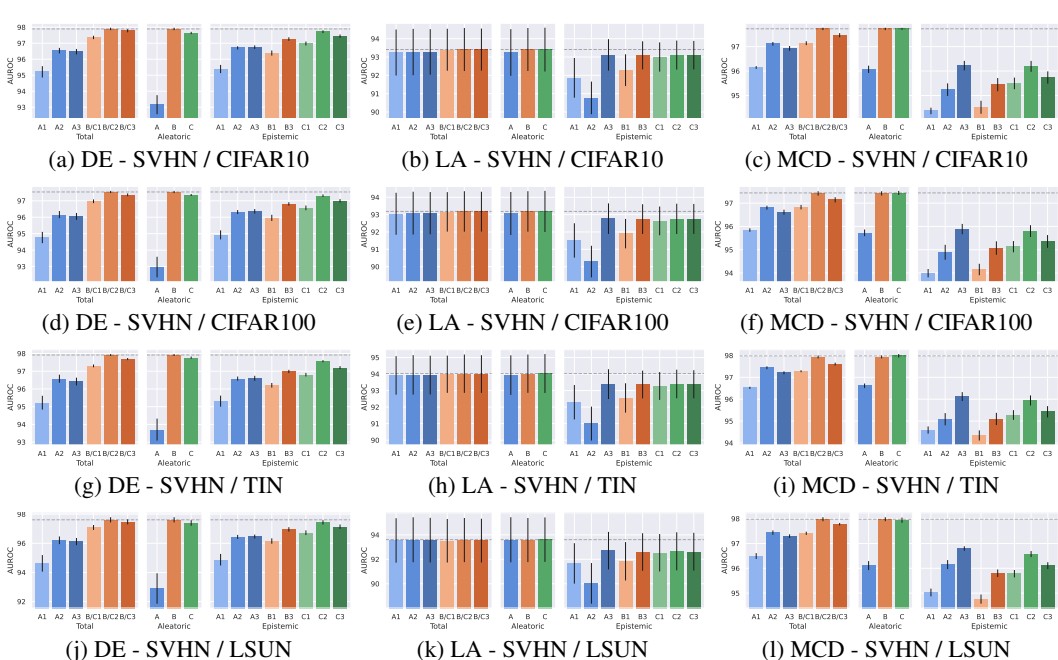

Figure 15: **OOD detection results for SVHN.** AUROC for distinguishing between ID and OOD datapoints using the different proposed measures of uncertainty as score. Means and standard deviations are calculated using five independent runs.

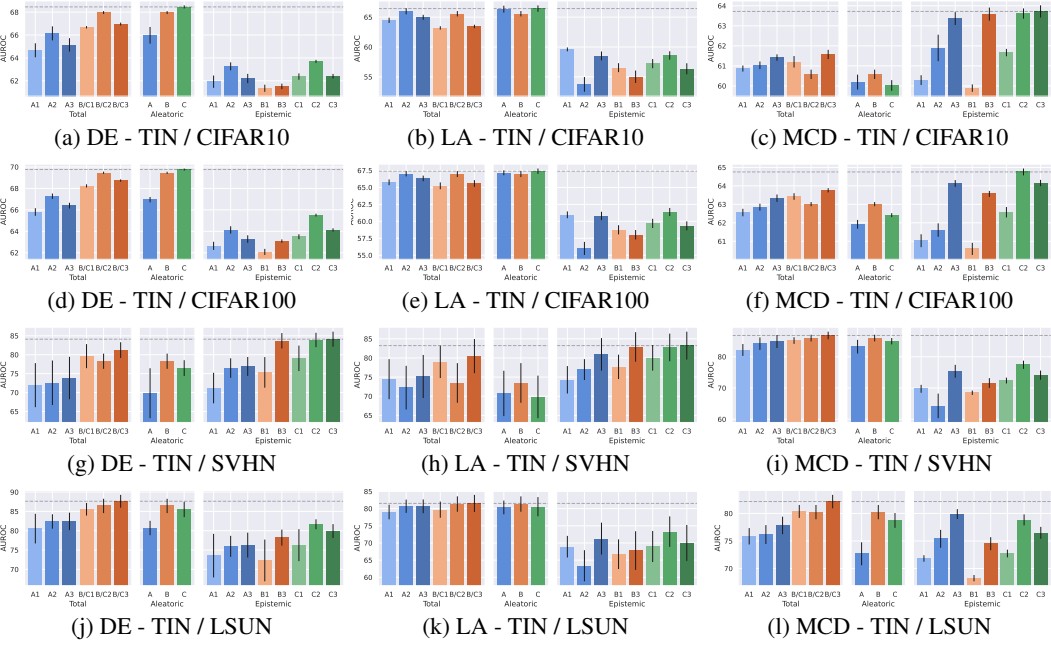

Figure 16: **OOD detection results for TIN.** AUROC for distinguishing between ID and OOD datapoints using the different proposed measures of uncertainty as score. Means and standard deviations are calculated using five independent runs.

## B.4 DIFFERENT NETWORK ARCHITECTURE

We want to assess the influence of the network architecture on the ranking of the results. To that end, we also trained DEs of DenseNet169 and RegNet-Y 800MF, using the same training recipe as for ResNet-18 described in Sec. 5. A comparison of the sampled models is given in Fig. 17. We observe, that ResNet-18 performs a bit better than the two other models, with RegNet-Y 800MF being the worst models in terms of NLL and accuracies. In terms of AU and EU, we observe only minor differences in the upper tails of the distributions for CIFAR100 and TIN. For CIFAR10 and SVHN, we observe no differences. Next, we analyze the influence of the network architecture on the misclassification and OOD detection tasks.

**Misclassification detection.** The results for misclassification detection using DEs with different model architectures are given in Fig. 18. We observe no major differences for different models (per column) under a given predicting model (per row).

**OOD detection.** The results for OOD detection using DEs with different model architectures are given in Fig. 19. We observe that the AU (C) is the best measure for DenseNet-169 and RegNet-Y 800MF, while it is AU (B) which is equivalent to TU (B/C2) for ResNet-18. However, the general trends are the same across all architectures.

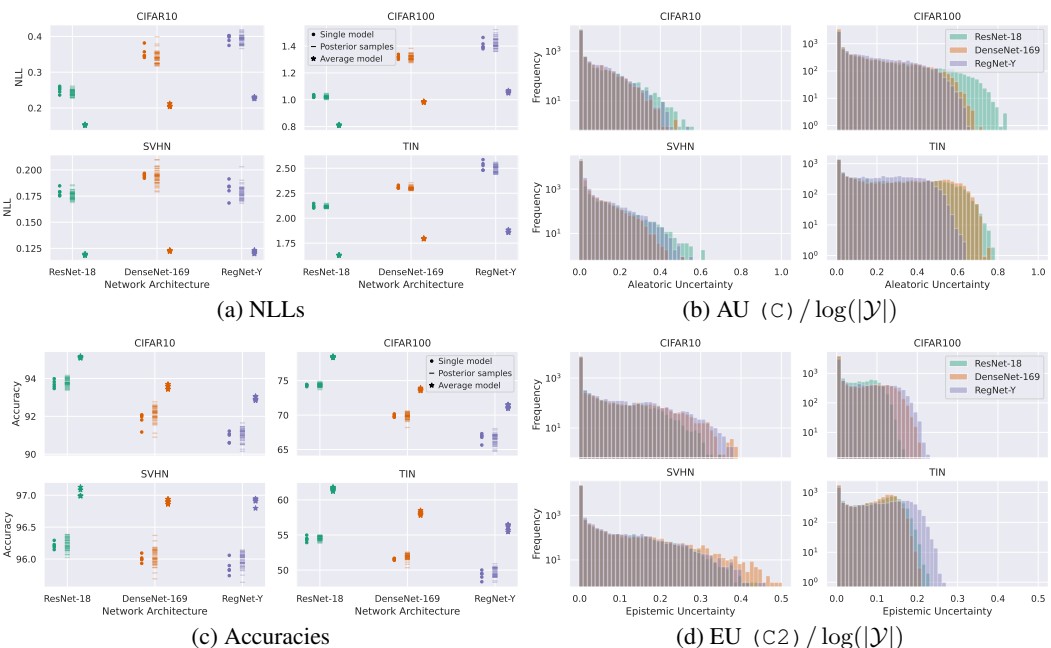

Figure 17: **Comparison of network architectures.** Results are obtained on the test split of the respective dataset. We compare the negative log-likelihoods (a) and accuracies (c) for different models obtained through DEs on ResNet-18, DenseNet-169 and RegNet-Y 800MF. The single model is randomly selected among all sampled models. We depict all models sampled in five independent runs. Furthermore, (b) the normalized AU (C) and (d) the normalized EU (C2) are given per sampling method. All three network architectures lead to similar results on all considered datasets.

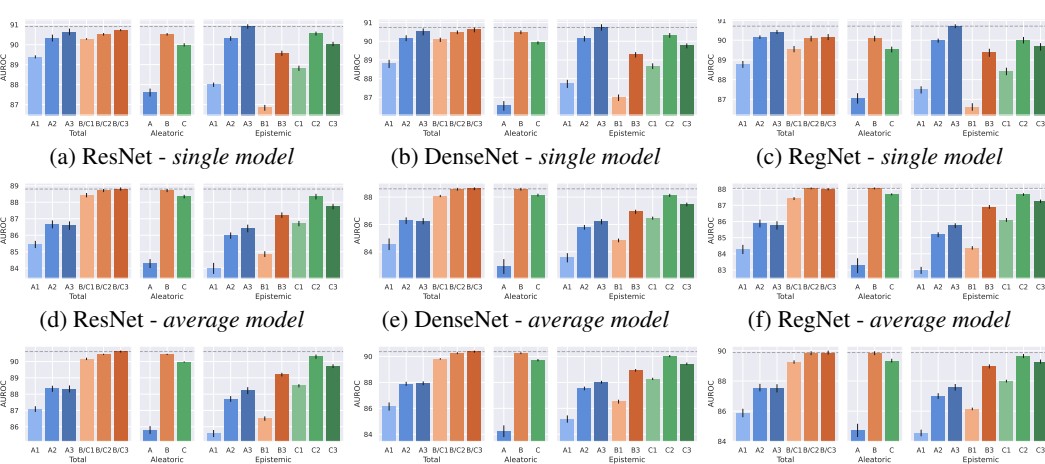

(a) ResNet - *single model*     (b) DenseNet - *single model*     (c) RegNet - *single model*

(d) ResNet - *average model*     (e) DenseNet - *average model*     (f) RegNet - *average model*

(g) ResNet - *according to posterior*   (h) DenseNet - *according to posterior*   (i) RegNet - *according to posterior*

Figure 18: **Misclassification detection results for DE with different model architectures and under different predicting models.** AUROC for distinguishing between correctly and incorrectly predicted samples under different predicting models, using the different proposed measures of uncertainty as score. Means and standard deviations are calculated using five independent runs.

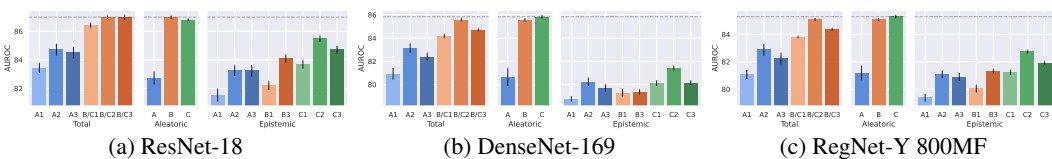

(a) ResNet-18        (b) DenseNet-169        (c) RegNet-Y 800MF

Figure 19: **OOD detection results for DE with different model architectures.** AUROC for distinguishing between ID and OOD datapoints using the different proposed measures of uncertainty as score. AUROCs are averaged over all ID / OOD combinations. Means and standard deviations are calculated using five independent runs.

## B.5 DISTRIBUTION SHIFT DETECTION

Next we want to assess the behavior of our framework of measures to detect varying levels of distribution shift. In this experiment, DE, LA and MCD are applied to CIFAR10 as training dataset. We use CIFAR10-C (Hendrycks and Dietterich, 2019) which contains corrupted versions of the test dataset of CIFAR10 to assess the performance of detecting distribution shifts. Therefore, we utilize the uncertainty as score to calculate the AUROC of distinguishing between the clean test dataset and the corrupted versions. We also investigated the AUPR and FPR@TPR95 as alternative metrics, which lead to equivalent conclusions. We utilized the 15 main corruptions and excluded the four additional corruptions intended for hyperparameter tuning by the authors of the dataset. Results are averages over all 15 corruptions. However, all corruptions are available in 5 different levels of severity, which we distinguish in our experiments.

The results in Fig. 20 show the AUROC of distinguishing between the clean and corrupted versions of the test dataset (y-axis) for different posterior sampling methods (rows), for different uncertainty measures (columns) under different corruption severities (x-axis). Furthermore, the inset shows a comparison akin to those done for OOD detection for the highest severity corrupted datapoints. We observe similar trends to those observed for the OOD detection experiments, which is not surprising given the similar nature of those experiments. However, comparing the best performing measure of uncertainty under DE for different severities shows, that EU is more effective than AU or TU at intermediate severities, but become equally effective for the highest severity. For LA, TU and AU measures all perform very similar across all severities. For MCD, we observe similar trends as for DE, albeit less pronounced.

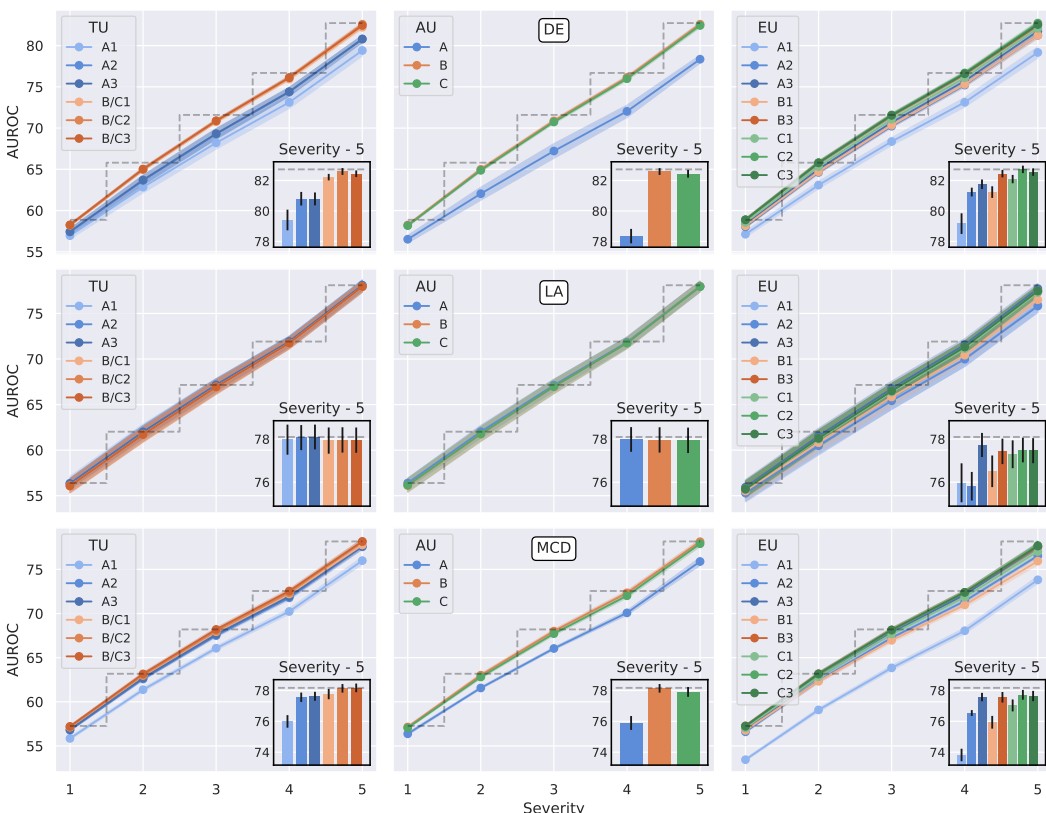

Figure 20: **Distribution shift detection on CIFAR10-C.** AUROC for distinguishing between clean and corrupted test datapoints, using the different proposed measures of uncertainty as score, under posterior sampling methods DE, LA and MCD. Black dashed line shows the maximum AUROC over all measures per severity. Insets show a comparison auf AUROCs uncer different uncertainty measures for the highest severity. Means and standard deviations are calculated using five independent runs.

## B.6 ADVERSARIAL EXAMPLE DETECTION

We want to investigate the effect of adversarially created inputs on the uncertainty estimates. Throughout this experiments, we consider adversarial attacks on the single network. However, it would also be possible to attack the average model, albeit more computationally expensive. As adversarial examples are known to transfer well between models of similar architecture (Goodfellow et al., 2015), results for attacking the average model are expected to be relatively similar to those presented here.

We consider two different adversarial attacks, (i) FGSM (Goodfellow et al., 2015) and (ii) PGD under infinity norm perturbation (Madry et al., 2018). For our experiments, we only consider the subset of the test datasets that are predicted correctly. This we refer to as the *original* dataset. Then we apply the adversarial attacks the datapoints in the original dataset and select those datapoints where the model was successfully fooled to predict incorrectly. This we refer to as the *adversarial* dataset. We utilize the different uncertainty scores to calculate the AUROC of distinguishing between the original and the adversarial dataset, akin to the OOD detection experiments reported in the main paper. We also investigated the AUPR and FPR@TPR95 as alternative metrics, which lead to equivalent conclusions.

**FGSM.** We start with the results obtained through the FGSM attack with $\epsilon = 8/255$. Histograms of the AU (A), the entropy of the predictive distribution of the single attacked model and the AU (B), the entropy of the predictive distribution under the average model, are shown in Fig. 21 for DE, in Fig. 22 for LA and in Fig. 23 for MCD. For all methods, we observe a shift towards higher AUs for the adversarial datapoints compared to the original datapoints. This effect is strongest for the global posterior sampling method DE, which is expected. Furthermore, the shift appears more pronounced for AU (B), which makes sense as the adversarial examples have been obtained with the single model.

The main results are shown in Fig. 24, denoting the AUROC of distinguishing between the original and the adversarial datapoints using the different measures of uncertainty as score. We observe qualitatively very similar results to the OOD detection experiments, in that TU and AU measures for cases (B) and (C) are the most effective. The same we observe for MCD, albeit less pronounced than for DE. For LA, all TU and AU measures perform basically on par. Surprisingly, EU measures underperform for adversarial example detection, irrespective of the posterior sampling method.

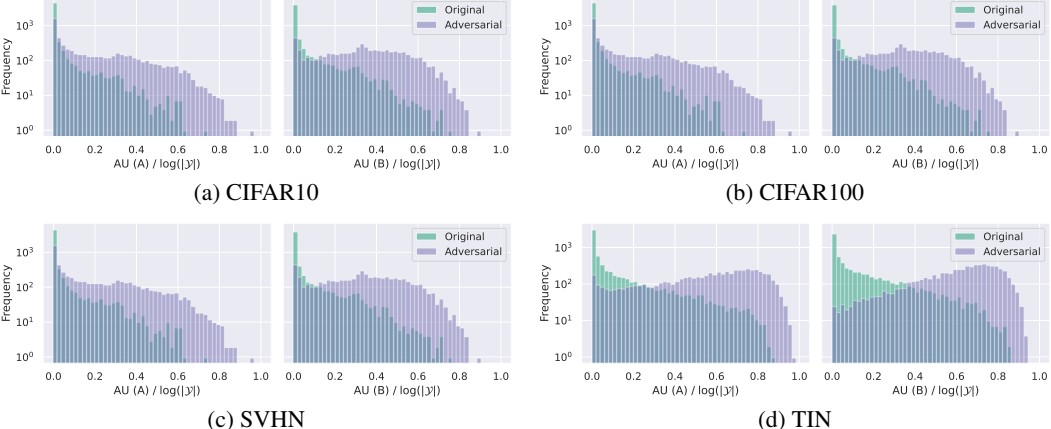

Figure 21: **Histogram of AU (A) and AU (B) for original and adversarial datapoints obtained through applying FGSM, using DE.** Aleatoric uncertainties are normalized with $\log(|\mathcal{Y}|)$ to be more comparable across datasets with different number of classes.

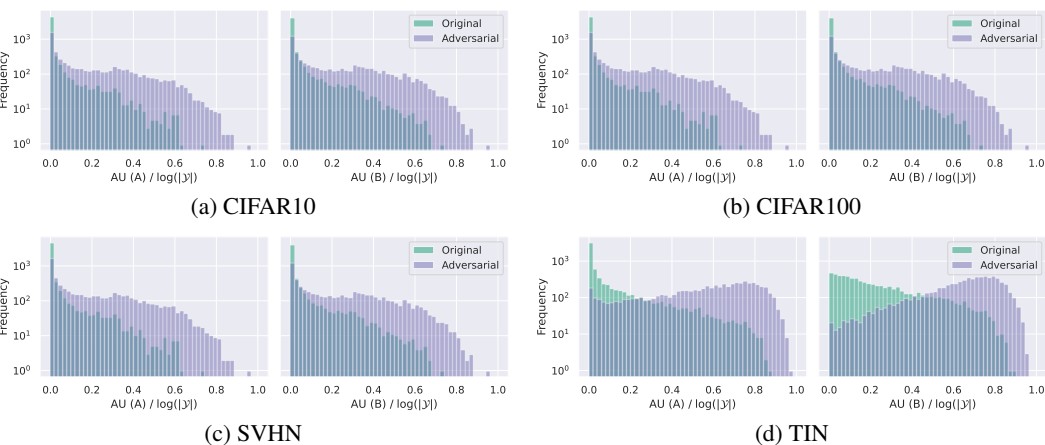

Figure 22: **Histogram of AU `(A)` and AU `(B)` for original and adversarial datapoints obtained through applying FGSM, using LA.** Aleatoric uncertainties are normalized with $\log(|\mathcal{Y}|)$ to be more comparable across datasets with different number of classes.

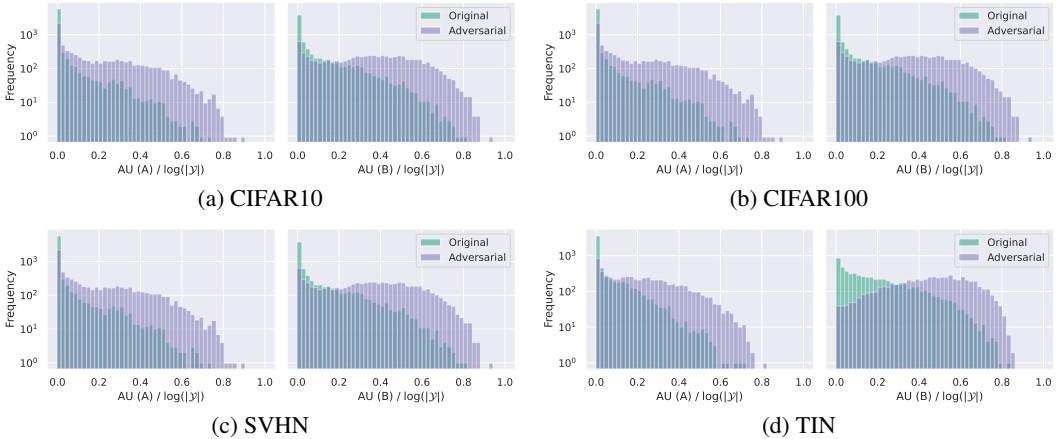

Figure 23: **Histogram of AU `(A)` and AU `(B)` for original and adversarial datapoints obtained through applying FGSM, using MCD.** Aleatoric uncertainties are normalized with $\log(|\mathcal{Y}|)$ to be more comparable across datasets with different number of classes.

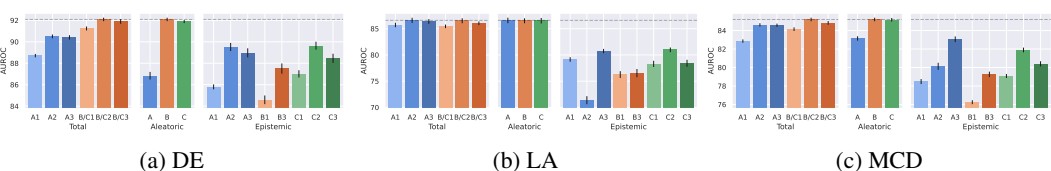

Figure 24: **Adversarial example detection (FGSM).** Means and standard deviations are calculated using five independent runs.

**PGD.** Next, we conduct the same investigation using the $L_\infty$-PGD attack with $\epsilon = 8/255$. Histograms of the AU (A), the entropy of the predictive distribution of the single attacked model and the AU (B), the entropy of the predictive distribution under the average model, are shown in Fig. 25 for DE, in Fig. 26 for LA and in Fig. 27 for MCD. For all methods, we observe a shift towards **lower** AUs for the adversarial datapoints compared to the original datapoints. The only exception is for AU (B) under DE, where adversarial datapoints exhibit slightly higher values than the original datapoints.

The results are given in Fig. 28, denoting the AUROC of distinguishing between the original and the adversarial datapoints using the different measures of uncertainty as score. For DE we observe that all measures except TU (A1) and AU (A) perform better than random. The very bad performance of AU (A) stems from the fact that adversarial datapoints exhibit lower uncertainties than the original datapoints (c.f. Fig. 25). The local posterior sampling methods LA and MCD exhibit worse than random performance for all considered measures of uncertainty. However, contrary to the experiments with PGD, measures of EU perform best.

The two experiments for adversarial example detection were conducted under the assumption that adversarial datapoints should exhibit higher uncertainty than the original datapoints. Finally, we investigate a special variant of our experiments with $L_\infty$-PGD adversarial examples, where we assume that adversarial datapoints exhibit lower uncertainty than the original datapoints. The results are shown in Fig. 29. We observe, that using AU (A) leads to the best results for all three posterior sampling methods. However this results do not help to attain a mechanism for adversarial robustness, as we leverage additional side information that the single model was fooled into being very confident about the adversarial examples. Attackers could add constraints on the deviation between the AU (A) under the original and the adversarial datapoint in an improved version of the $L_\infty$-PGD attack, rendering this detection mechanism useless.

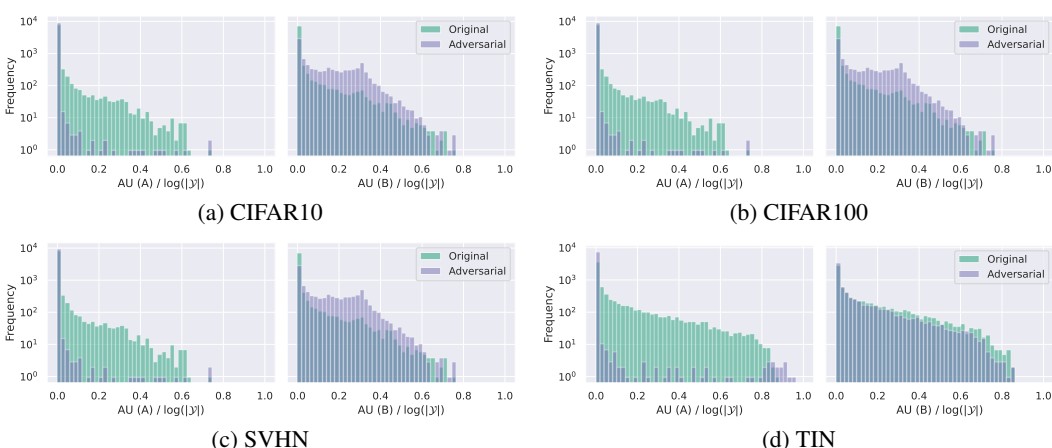

(a) CIFAR10      (b) CIFAR100

(c) SVHN      (d) TIN

Figure 25: **Histogram of AU (A) and AU (B) for original and adversarial datapoints obtained through applying $L_\infty$-PGD, using DE.** Aleatoric uncertainties are normalized with $\log(|\mathcal{Y}|)$ to be more comparable across datasets with different number of classes.

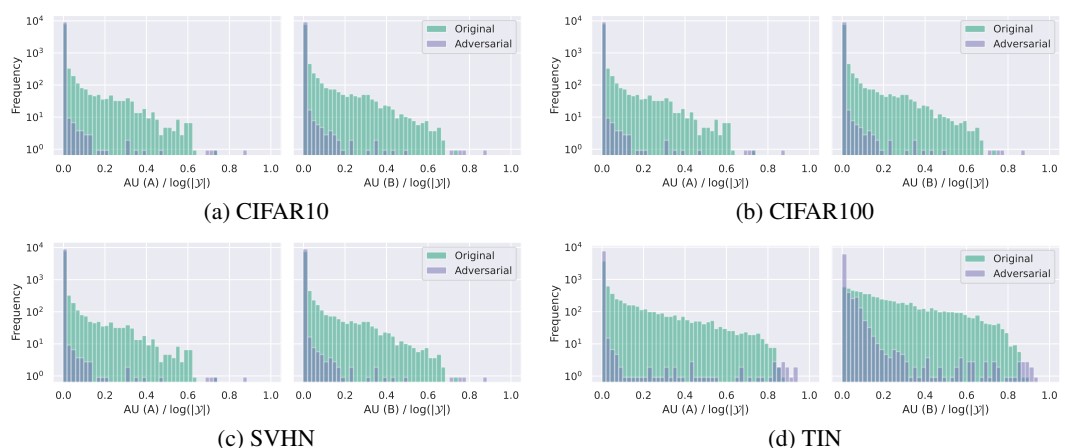

Figure 26: **Histogram of AU (A) and AU (B) for original and adversarial datapoints obtained through applying $L_\infty$-PGD, using LA.** Aleatoric uncertainties are normalized with $\log(|\mathcal{Y}|)$ to be more comparable across datasets with different number of classes.

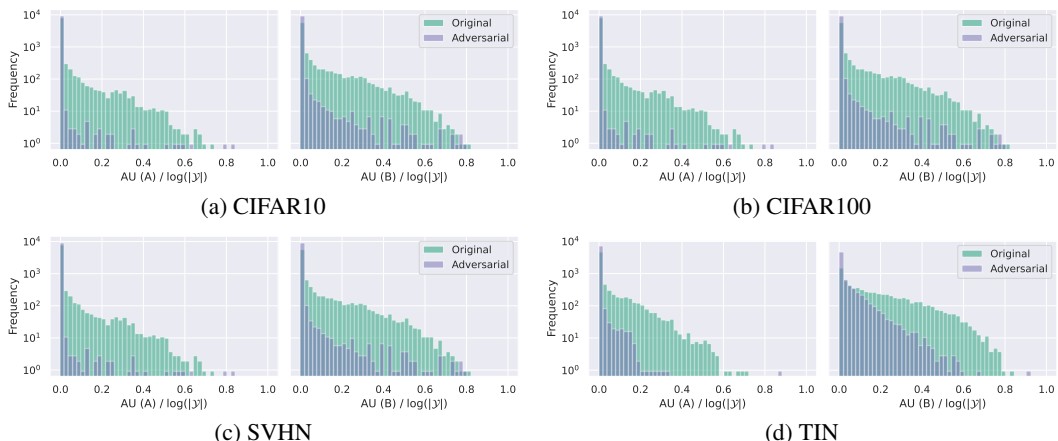

Figure 27: **Histogram of AU (A) and AU (B) for original and adversarial datapoints obtained through applying $L_\infty$-PGD, using MCD.** Aleatoric uncertainties are normalized with $\log(|\mathcal{Y}|)$ to be more comparable across datasets with different number of classes.

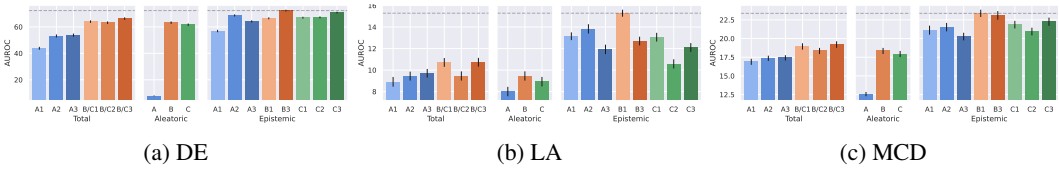

Figure 28: **Adversarial example detection ($L_\infty$-PGD).** Means and standard deviations are calculated using five independent runs.

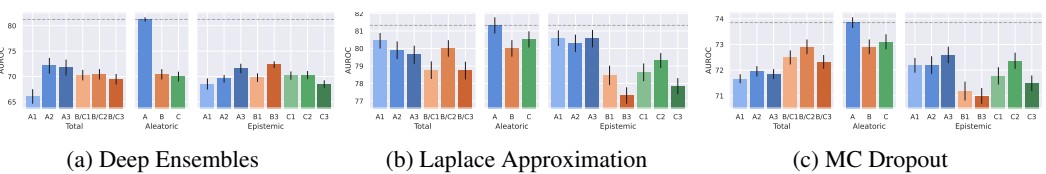

Figure 29: **Adversarial example detection ($L_\infty$-PGD), switching clean and adversarial dataset.** Means and standard deviations are calculated using five independent runs.

## B.7 ACTIVE LEARNING

Finally, we investigated the proposed framework of uncertainty measures on active learning tasks. We conducted experiments on the MNIST and FMNIST datasets. A small CNN (5x5 conv [1 to 6 channels], 2x2 max-pool, 5x5 conv [6 to 12 channels], 2x2 max-pool, two linear layers with hidden size 32 and a final output linear layer; ReLU activations after each max-pool and linear layer except the last as well as dropout with dropout between linear layers) was utilized. For DE, the dropout rate was set to zero, for MCD it was set to 0.2 for experiments on both datasets. Training of the models utilized the Adam optimizer (Kingma and Ba, 2015) for 50 epochs with a learning rate of 1e-3, a batch size of 32 and l2 weight decay of 1e-4. Early stopping was performed on the official validation split of the respective datasets, the evaluation of the performance per step was conducted on the official test splits. Note that even though the size of the training dataset increases each step, the effective size, thus the number of gradient steps per epoch, was kept constant at 1000 for the MNIST and 1600 for the FMNIST experiments. For DE, we obtain 5 posterior samples (ensemble members), for MCD we obtain 50 posterior samples. The average over those samples, the approximated posterior predictive, was used to calculate the accuracies for each acquisition step, as well as for selecting the next datapoints to add to the training dataset from the pool dataset.

**MNIST.** We started with 20 datapoints in the training dataset and the remaining 49,980 datapoints in the pool dataset. Those 20 datapoints were balanced, such that two datapoints from each class were contained. Each iteration, the five samples with the highest uncertainty are transferred from the pool dataset to the training dataset. We considered TU, AU and EU for measures (B2), (B3), (C2) and (C3) as acquisition functions, as well as random selection as a baseline. We did not investigate measures (A1), (A2), (A3), (B1) and (C1) due to the long runtimes of the experiments, but would expect them to perform worse than the considered ones in light of the other experiments we conducted. An interesting situation could be the EU (A1) however, when training a single model on the dataset in the current iteration and compare the model from the previous iteration. Future work should investigate this setting, e.g. in transfer learning settings.

The results are given in Fig. 30. We observe, that for both DE as well as MCD, EU (C2), the mutual information, leads to the best performance at the final iteration, as well as performs very well throughout all iterations. Interestingly, we find TU (B/C2) which is identical to AU (B) to be equally well performing for both cases. The same is found for TU (B/C3). Interestingly, the EU (B3) and EU (C3) are the worst performing acquisition functions for both DE and MCD, contrary to the sentiment that estimators of EU should perform best in this task. Similarly surprising, AU (C), which is an asymptotically unbiased estimator of the aleatoric uncertainty of the true model, performs very good as acquisition function for DE. It is the worst acquisition function though for MCD. The random sampling baseline is also extremely effective until around a training dataset size of around 100 samples, more effective than any of the considered uncertainty measures. We hypothesize, that until a certain dataset size, models sampled from the posterior are not specified enough and provide too little signal of what datapoints to add next, which would be interesting to investigate in more details in future experiments.

**FMNIST.** We started with 1000 datapoints in the training dataset and the remaining 49,000 datapoints in the pool dataset. Those 1000 datapoints were balanced, such that 100 datapoints from each class were contained. Each iteration, the 15 samples with the highest uncertainty are transferred from the pool dataset to the training dataset. As for the MNIST experiment, we considered TU, AU and EU for measures (B2), (B3), (C2) and (C3) as acquisition functions, as well as random selection as a baseline.

The results are provided in Fig. 31. For MCD, we do not see a clear trend of outperforming the random acquisition baseline with any uncertainty measure. For DE, we again observe that EU (C2), the mutual information, leads to very good performance throughout all acquisition steps. Also, TU (B/C2) which is identical to AU (B) and AU (C) perform very good. Again, EU (B3) and EU (C3) are the worst performing acquisition functions, especially towards the final steps. This seemingly similar task to MNIST proved to be surprisingly difficult for an active learning pipeline, potentially due to the higher difficulty of the task where class boundaries are known to be much harder to learn.

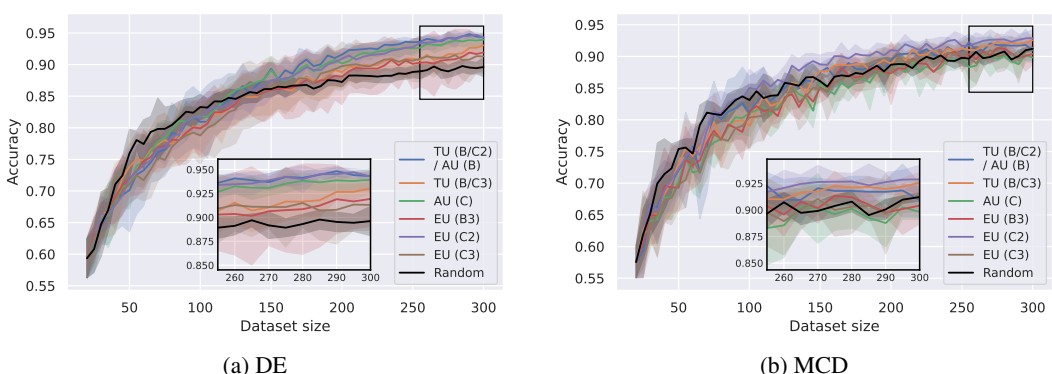

(a) DE                    (b) MCD

Figure 30: **Active learning results on MNIST.** TU, AU and EU for measures `(B2)`, `(B3)`, `(C2)` and `(C3)` were considered as acquisition functions. The accuracy is those of the average model. Means and standard deviations are calculated using five independent runs.

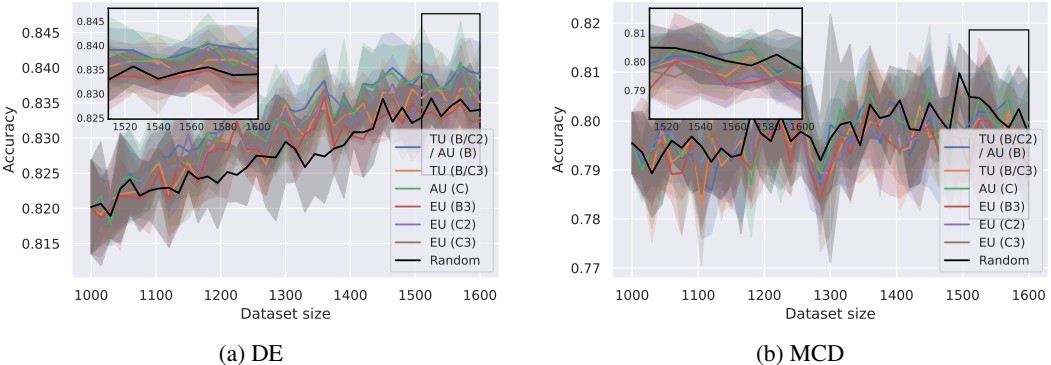

(a) DE                    (b) MCD

Figure 31: **Active learning results on FMNIST.** TU, AU and EU for measures `(B2)`, `(B3)`, `(C2)` and `(C3)` were considered as acquisition functions. The accuracy is those of the average model. Means and standard deviations are calculated using five independent runs.

